# **AdaPTS**: Adapting Univariate Foundation Models to Probabilistic Multivariate Time Series Forecasting

Abdelhakim Benechehab [1][2]   Vasilii Feofanov [1]   Giuseppe Paolo [1]   Albert Thomas [1]
Maurizio Filippone [3]   Balázs Kégl [1]

## Abstract

Pre-trained foundation models (FMs) have shown exceptional performance in univariate time series forecasting tasks. However, several practical challenges persist, including managing intricate dependencies among features and quantifying uncertainty in predictions. This study aims to tackle these critical limitations by introducing **adapters**—feature-space transformations that facilitate the effective use of pre-trained univariate time series FMs for multivariate tasks. Adapters operate by projecting multivariate inputs into a suitable latent space and applying the FM independently to each dimension. Inspired by the literature on representation learning and partially stochastic Bayesian neural networks, we present a range of adapters and optimization/inference strategies. Experiments conducted on both synthetic and real-world datasets confirm the efficacy of adapters, demonstrating substantial enhancements in forecasting accuracy and uncertainty quantification compared to baseline methods. Our framework, **AdaPTS**, positions adapters as a modular, scalable, and effective solution for leveraging time series FMs in multivariate contexts, thereby promoting their wider adoption in real-world applications. We release the code at https://github.com/abenechehab/AdaPTS.

## 1. Introduction

Time series forecasting is a well-established machine learning task that involves analyzing sequential data to predict future trends based on historical patterns. Two key challenges frequently arise in this context: (a) time series are

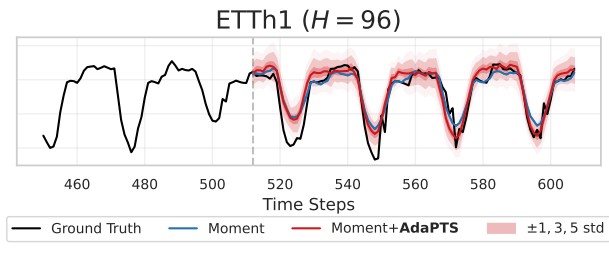

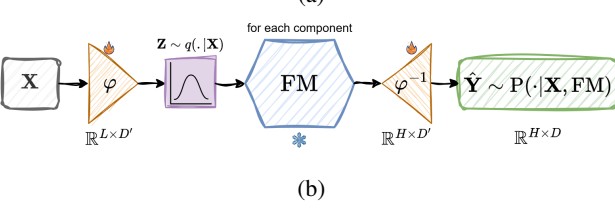

Figure 1: (a) Augmenting `Moment` time series foundation model with the **AdaPTS** framework provides *probabilistic* and *more accurate* predictions. (b) **The AdaPTS framework:** The input time series is transformed through a feature space transformation $\varphi$ that maps into a stochastic latent space, where the FM now operates. The fire symbol indicate trainable weights while the snowflake implicates that the parameters of the FM are kept frozen.

often multivariate, incorporating multiple descriptive features (Wei, 2019), and (b) estimating the uncertainty of a forecast is equally important, requiring probabilistic model outputs (Gneiting & Katzfuss, 2014). These challenges are particularly relevant in real-world applications where risk assessment depends on reliable forecasts, such as healthcare (Jones & Spiegelhalter, 2012), finance (Groen et al., 2013), energy management (Zhang et al., 2014; Nowotarski & Weron, 2018), and weather prediction (Palmer, 2012; Bi et al., 2023).

Existing foundation models (FMs) for time series forecasting, such as `Chronos` (Ansari et al., 2024), are typically trained for univariate forecasting tasks due to tractability constraints, as a wide range of real-world time series problems often involve different numbers of features. Nevertheless, handling multivariate time series directly within these

[1]Huawei Noah's Ark Lab, Paris [2]Department of Data Science, EURECOM [3]Statistics Program, KAUST. Correspondence to: Abdelhakim Benechehab <abdelhakim.benechehab@gmail.com>.

*Proceedings of the $42^{nd}$ International Conference on Machine Learning*, Vancouver, Canada. PMLR 267, 2025. Copyright 2025 by the author(s).

models has been attempted (Liu et al., 2024), yet remains computationally challenging due to the high-dimensional dependencies among features. This limitation raises a fundamental question: how can we leverage existing pre-trained univariate FMs to enable probabilistic forecasting for multivariate time series?

To address this, we introduce **AdaPTS**, a novel framework designed to augment FMs with probabilistic adapters. As illustrated in Figure 1, **AdaPTS** applies a stochastic feature transformation that maps the input time series into a latent space, where predictions are made using the frozen FM. Our framework sets itself apart from existing literature by fitting an inverse transformation as part of the adapter, allowing predictions to be transformed back into the original feature space. Beyond enhancing forecasting accuracy, the integration of stochasticity into the adapter's latent representation ensures that the model captures uncertainty, thereby improving both calibration and robustness.

Our approach leads to several novel insights and contributions, which we summarize as follows:

1. **Multivariate FM adaptation.** We introduce a principled methodology for adapting existing pre-trained univariate FMs to multivariate probabilistic forecasting, resulting in the **AdaPTS** framework.

2. **Theoretical foundations of adapters.** We provide a theoretical analysis to support the necessity of adapters, starting with the analytically tractable case of linear adapters and linear FMs. We then build on the literature on partially stochastic Bayesian neural networks to introduce probabilistic adapters.

3. **Empirical validation.** We conduct experiments on multivariate time series forecasting benchmarks, demonstrating that our approach improves forecasting accuracy over baseline methods. We then analyze the interpretability of the learned latent representation and show that adapters enable cost-efficient adaptation by reducing the dimensionality of the feature space.

The rest of this paper is organized as follows: Section 2 states the problem setup and related work, Section 3 details the theoretical analysis on linear adapters. Section 4 extends our framework to probabilistic adapters, while Section 5 showcases experimental results. Finally, we conclude with the accompanying Python package in Section 6, and future directions in Section 7.

## 2. Background knowledge

In this section, we first define rigorously the problem, and then discuss the related work.

### 2.1. Problem setup

Consider a multivariate long-term time series forecasting task, represented by: a data matrix $\mathbf{X} \in \mathbb{R}^{L \times D}$ where $L$ is the context window size and $D$ is the multivariate time series dimensionality, and a target matrix $\mathbf{Y} \in \mathbb{R}^{H \times D}$, where $H$ is the forecasting horizon. We denote by $\mathbf{x}_d \in \mathbb{R}^{L \times 1}$ (respectively $\mathbf{y}_d \in \mathbb{R}^{H \times 1}$) the $d$-th component of the input (respectively target) multivariate time series.

Our goal is to leverage a frozen pre-trained univariate time series foundation model denoted by $f_{\text{FM}} : \mathbb{R}^{L \times 1} \to \mathbb{R}^{H \times 1}$ (Fig. 1b). In the case of a multivariate input $\mathbf{X}$, for the sake of simplicity, we denote by $f_{\text{FM}}(\mathbf{X})$ the application of $f_{\text{FM}}$ to each channel independently. We aim to achieve best forecasting quality by minimizing the mean squared error (MSE) loss defined as follows:

$$\mathcal{L} = \|\mathbf{Y} - f_{\text{FM}}(\mathbf{X})\|_{\text{F}}^2 = \frac{1}{D} \sum_{d=1}^{D} \|\mathbf{y}_d - f_{\text{FM}}(\mathbf{x}_d)\|_2^2. \quad (1)$$

We now formally define an adapter, a tool by means of which we aim to best use the foundation model $f_{\text{FM}}$ for multivariate forecasting:

**Definition 2.1** (adapter). An adapter is a feature-space transformation $\varphi : \mathbb{R}^D \to \mathbb{R}^{D'}$ that is applied to the data prior to the foundation model[1]. The forecast is then obtained by transforming the predictions back to the original feature space:

$$\hat{\mathbf{Y}}(\mathbf{X}; \varphi) = \varphi^{-1}\big(f_{\text{FM}}(\varphi(\mathbf{X}))\big).$$

According to this definition, an adapter is valid only if the inverse transformation $\varphi^{-1} : \mathbb{R}^{D'} \to \mathbb{R}^D$, such that $\forall \mathbf{x} \in \mathbb{R}^D$, $\varphi^{-1} \circ \varphi(\mathbf{x}) = \mathbf{x}$, is well-defined on $\mathbb{R}^{D'}$. In the rest of the paper, we relax this condition by implementing the direct transformation as *encoder* ($\varphi \triangleq \text{enc}$), and respectively, the inverse transformation as *decoder* ($\varphi^{-1} \triangleq \text{dec}$). In this case, the predictions obtained after the application of the adapter become: $\hat{\mathbf{Y}}(\mathbf{X}; \text{enc}, \text{dec}) = \text{dec}\big(f_{\text{FM}}(\text{enc}(\mathbf{X}))\big)$.

We note that in the literature, there exist alternatives to adapt to the multivariate setting (Zhang & Yan, 2023; Tian et al., 2023), but we have chosen this family of adapters due to their high flexibility as: (a) any foundation model can be plugged-in, (b) no requirement of fine-tuning due to feature-level transformations (Feofanov et al., 2024), (c) adaptation to the computation budget by defining the number of encoded channels.

**Optimality of an adapter.** In order for an adapter to be useful, it has to achieve a forecasting error lower compared to an adapter that represents the identity matrix $\mathbf{I}$ of size $D$ and does not perform any transformation of the feature

---

[1]In practice, $\varphi$ is applied on matrices $\mathbf{X}$ in $\mathbb{R}^{L \times D}$. This denotes the application of $\varphi$ on each row of $\mathbf{X}$.

space. Therefore, we define the optimality of the adapter based on improving the forecasting error of the identity baseline:

$$\mathcal{L} \geq \mathcal{L}(\varphi) = \|\mathbf{Y} - \varphi^{-1}\big(f_{\mathrm{FM}}(\varphi(\mathbf{X}))\big)\|_{\mathrm{F}}^2,$$

where $\mathcal{L}$ is defined by Eq. (1).

## 2.2. Related work

**Time Series Foundational Models.** Over the past two years, a plethora of foundation models have been proposed with a particular focus on time series forecasting. Some of these models like `GPT4TS` (Tian et al., 2023) and `Time-LLM` (Jin et al., 2024) "reprogram" a Large Language Model to the forecasting setting by freezing most of its layers and fine-tuning additional time series-specific modules to a new downstream task. The majority of time series FMs including `Lag-Llama` (Rasul et al., 2024), `Chronos` (Ansari et al., 2024), `Moirai` (Liu et al., 2024), `TimesFM` (Das et al., 2024) and `Moment` (Goswami et al., 2024) are trained from scratch on a large volume of time series data. Beyond the transformer architecture, Ekambaram et al. (2024) present `TTM`, an all MLP foundation model.

**Adapters.** The multivariate setting presents a significant challenge for time series FMs, as different tasks involve varying numbers of channels[2]. To the best of our knowledge, the only model that naturally accommodates any number of channels is `Moirai` (Liu et al., 2024), which, however, suffers from high computational demand due to processing all channels flattened in the transformer, leading to a quadratic memory complexity w.r.t. to the number of channels. Most foundation models, instead, treat each one of these independently, which, as noted by Feofanov et al. (2024; 2025), remains computationally expensive when full fine-tuning is required. For classification tasks with hundreds or thousands of features, they demonstrated that simple adapters like the rotation matrix obtained through Principal Components Analysis (`PCA`) mitigate this issue. At the same time, Benechehab et al. (2025) showed that `PCA` preserves channel interactions by learning new disentangled components. However, in both cases, `PCA` provided little improvement over independent processing, leaving room for further enhancements. In the context of tabular regression, foundation models such as `TabDPT` (Ma et al., 2024) also use `PCA` to adapt to a variable number of features.
Less related to our work, Li & Marlin (2016) use a Gaussian process adapter in the context of irregular time series classification. In other domains, adapters have been used for multimodal (text-time series) representation learning (Zhang et al., 2024) and computer vision (Li et al., 2025; Yin et al., 2023; Pan et al., 2022).

---

[2]Throughout the paper, the words *features*, *channels*, and *components* are used interchangeably to refer to the number of variates in a multivariate time series, represented as $D$ in Section 2.1.

## 3. Theoretical analysis

The purpose of this section is to study the optimization problem that the adapter $\varphi$ is aiming to solve:

$$\varphi^* = \arg\min_{\varphi} \|\mathbf{Y} - \varphi^{-1}\big(f_{\mathrm{FM}}(\varphi(\mathbf{X}))\big)\|_{\mathrm{F}}^2. \quad (2)$$

Under mild assumptions on the adapter function class and the backbone foundation model $f_{\mathrm{FM}}$, we aim at characterizing the optimal solution $\varphi^*$ and prove that it realizes the optimality condition: $\mathcal{L}(\varphi^*) \leq \mathcal{L}$.

We first consider the linear case where we constrain the adapter $\varphi$ to the class of linear transformations, parametrized by a matrix $\mathbf{W}_\varphi \in \mathbb{R}^{D \times D}$: $\varphi(\mathbf{X}) = \mathbf{X}\mathbf{W}_\varphi$.

**Assumption 3.1.** $\mathbf{W}_\varphi$ has full rank: $\mathrm{rank}(\mathbf{W}_\varphi) = D$, insuring its invertibility.

**Assumption 3.2.** For ease of derivation, we consider a similar linear parametrization for the foundation model: $f_{\mathrm{FM}}(\mathbf{X}) = \mathbf{W}_{\mathrm{FM}}^\top \mathbf{X} + \mathbf{b}_{\mathrm{FM}} \mathbf{1}^\top$ where $\mathbf{W}_{\mathrm{FM}} \in \mathbb{R}^{L \times H}$, $\mathbf{b}_{\mathrm{FM}} \in \mathbb{R}^H$, and $\mathbf{1}$ a vector of ones of dimension $D$.

**Proposition 3.3** (Optimal linear adapter). *Under Assumption 3.1 and Assumption 3.2, the closed-form solution of the problem*

$$\mathcal{L}(\mathbf{W}_\varphi) = \|\mathbf{Y} - \big(\mathbf{W}_{FM}^\top \mathbf{X}\mathbf{W}_\varphi + \mathbf{b}_{FM}\mathbf{1}^\top\big)\mathbf{W}_\varphi^{-1}\|_F^2 \quad (3)$$

*writes as:*

$$\mathbf{W}_\varphi^* = (\mathbf{B}^\top \mathbf{A})^+ \mathbf{B}^\top \mathbf{B}, \quad (4)$$

*where $\mathbf{W}_\varphi^* = \arg\min_{\mathbf{W}_\varphi} \mathcal{L}(\mathbf{W}_\varphi)$, $\mathbf{A} = \mathbf{Y} - \mathbf{W}_{FM}^\top \mathbf{X}$, $\mathbf{B} = \mathbf{b}_{FM}\mathbf{1}^\top$, and $(\mathbf{B}^\top \mathbf{A})^+$ denoting the pseudo-inverse operator.*

*Proof.* The result follows by differentiating $\mathcal{L}(\mathbf{W}_\varphi)$ with respect to $\mathbf{W}_\varphi$, and solving the Euler equation: $\nabla_{\mathbf{W}_\varphi}\mathcal{L} = 0$. The detailed proof is deferred to Appendix A. $\square$

*Remark* 3.4. In this case, the fact that the matrix $\mathbf{B} = \mathbf{b}_{FM}\mathbf{1}^\top$ have identical columns renders the matrix $\mathbf{B}^\top\mathbf{A}$ degenerate (with $rank(\mathbf{B}^\top\mathbf{A}) = 1$). In practice, we add a positive constant to the diagonal in order to numerically stabilize the matrix inversion: $\mathbf{W}_\varphi^* = (\mathbf{B}^\top\mathbf{A} + \lambda\mathbf{I})^{-1}\mathbf{B}^\top\mathbf{B}$, with $\lambda > 0$. In Section 3.1, we show that we are able to reach an optimal solution regardless of this added regularization.

Proposition 3.3 establishes the existence of a solution to the optimal adapter problem defined in Eq. (2) and demonstrates that this solution does not necessarily correspond to the identity matrix. This analysis in the linear case provides the foundational motivation for our gradient descent based optimization approach to determine the optimal adapter configuration.

### 3.1. Working example

**Synthetic data.** Our synthetic dataset comprises two multivariate time series, one with several independent components and the other with linearly correlated ones (Fig. 2), designed to evaluate a linear feature-space transformation. The data generation process creates five *uncorrelated* base signals (sinusoids with distinct frequencies, amplitudes, and iid noise) and derives eight additional channels through linear combinations of these bases with additive Gaussian noise of different magnitude $\sigma \in (0.1, 0.2, 0.5)$.

This construction provides a controlled environment where the ground truth relationship between channels is known: the underlying data manifold is effectively five-dimensional, but in the correlated case the observed eight-dimensional multivariate time series includes varying levels of noise and linear mixing.

**Randomly generated linear FMs.** The experimental setup in Fig. 2 consists in randomly sampling the linear parameters of a toy foundation model: $\mathbf{W}_{\text{FM}}$ and $b_{\text{FM}}$. To simulate a realistic scenario, we use Glorot-uniform initialization distribution as it would be the case in neural network-based architectures. We then compute the closed-form solution $\mathbf{W}_\varphi^*$ Eq. (4) on raw data $\mathbf{X}$, and compare the resulting loss value with the baseline (using the identity matrix $\mathbf{I}$ as adapter) and the PCA-only adapter.

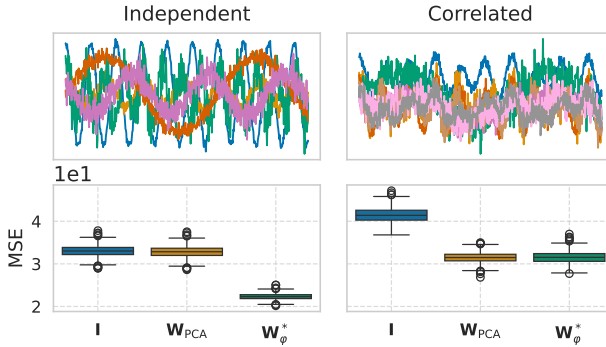

Figure 2: **Optimality of $\mathbf{W}_\varphi^*$.** Comparing the MSE obtained with $\mathbf{W}_\varphi^*$ against the baseline, for 1000 randomly generated linear FM.

Fig. 2 shows that in the case of uncorrelated data (*left* column) PCA is equivalent to the identity matrix, while the solution $\mathbf{W}_\varphi^*$ to the problem $\mathcal{L}(\mathbf{W}_\varphi)$ reaches an order of magnitude better forecasting loss. In the correlated case, we observe that PCA has a similar performance to the optimal solution. This example motivates the adapter idea through the existence of better linear transformations than the identity matrix in the case of linear foundation models.

## 4. **AdaPTS**: Adapters for Probabilistic Multivariate Time Series Forecasting

In this section, we introduce families of adapters that verify the conditions stated in Definition 2.1. Furthermore, we extend this definition to include probabilistic variants of adapters, useful for uncertainty quantification.

### 4.1. Families of adapters

Our framework accommodates various families of transformations that can serve as adapters. Initially, we define linear AutoEncoders and subsequently extend them to their deep non-linear counterparts. Ultimately, we introduce the probabilistic adapter framework, encompassing Variational AutoEncoders (VAE) and Dropout-based families of adapters.

**Linear AutoEncoders.** In addition to the linear setup introduced in Section 3, we extend Linear AutoEncoders to provide a simple yet effective method for dimensionality reduction while preserving the temporal relationships within time series data. In this more general case, the encoder compresses the multivariate time series $\mathbf{X}$ into a potentially lower-dimensional representation $\mathbf{Z} = \mathbf{X}\mathbf{W}_{\text{enc}}$, where $\mathbf{W}_{\text{enc}} \in \mathbb{R}^{D \times D'}$ is the linear transformation matrix, and $D' \leq D$. The decoder reconstructs the forecast to the original feature space as $\hat{\mathbf{Y}} = f_{\text{FM}}(\mathbf{Z})\mathbf{W}_{\text{dec}}$. Finally, the parameters of the encoder $\mathbf{W}_{\text{enc}}$ and the decoder $\mathbf{W}_{\text{dec}}$ are jointly optimized to minimize the objective in Eq. (2).

**Deep non-linear AutoEncoders.** Deep non-linear AutoEncoders extend their linear counterparts by employing multiple layers of non-linear transformations. The encoder maps the input $\mathbf{X}$ to a latent space $\mathbf{Z} = \text{enc}(\mathbf{X}; \theta_{\text{enc}})$, where enc is parameterized by a deep neural network. Similarly, the decoder reconstructs the predictions of the foundation model in the latent space: $\hat{\mathbf{Y}} = \text{dec}(f_{\text{FM}}(\mathbf{Z}); \theta_{\text{dec}})$.

Besides AutoEncoders, Normalizing Flows (Kobyzev et al., 2021) such as RealNVP (Dinh et al., 2017) are a valid choice in the context of adapters, thanks to their inherently invertible nature. However, their training proved challenging due to the transformation being tied to its inverse leading to poor performance in practice. We defer a discussion on Normalizing Flows as adapters to Appendix B.

### 4.2. Probabilistic Adapters

We now discuss an alternative to the optimization of adapters, which is based on a Bayesian treatment of their parameters. There are many options on how to carry out inference over these parameters, and we can draw from the literature on Bayesian inference for neural networks (Papamarkou et al., 2024).

Considering a FM which yields point predictions, the appeal

of Bayesian adapters is that they enable probabilistic predictions, which can be used for uncertainty quantification. Note that this is the case for models such as Chronos and Moirai, which output a distribution over the time series continuous values[3]. For deterministic FMs such as Moment (Goswami et al., 2024), a Bayesian treatment of adapters yields an ensemble of such predictions, which is key for accounting for the predictive uncertainty.

**Inference.** Recalling that $\theta$ represents the set of parameters of encoder ($\text{enc}_\theta$) and decoder ($\text{dec}_\theta$), we can attempt to obtain the posterior distribution over these parameters through Bayes theorem (Gelman et al., 2013):

$$p(\theta|\mathbf{Y}, \mathbf{X}) \propto p(\mathbf{Y}|\mathbf{X}, \theta)p(\theta),$$

where $p(\theta)$ is the prior distribution over the parameters and $p(\mathbf{Y}|\mathbf{X}, \theta)$ the likelihood, with $\mathbf{Y}, \mathbf{X}$ representing a training dataset in this context. Alternatively, we can treat the latent representation $\mathbf{Z}$ as stochastic, where the interest is now to characterize the following posterior:

$$p(\mathbf{Z}|\mathbf{Y}, \mathbf{X}) \propto p(\mathbf{Y}|\mathbf{X}, \mathbf{Z})p(\mathbf{Z}).$$

In these two formulations, the posterior distribution over the parameters is instrumental in obtaining predictive distributions useful for uncertainty quantification. For instance, in the case of inference over $\theta$, for new test data $\mathbf{Y}^*, \mathbf{X}^*$ we obtain:

$$p(\mathbf{Y}^*|\mathbf{X}^*, \mathbf{Y}, \mathbf{X}) = \int p(\mathbf{Y}^*|\mathbf{X}^*, \theta)p(\theta|\mathbf{Y}, \mathbf{X})d\theta.$$

Characterizing the posterior analytically, however, is intractable and we need to resort to approximations. The literature on Bayesian inference offers various strategies, which can be adapted to neural networks (Papamarkou et al., 2024), including variational inference (Graves, 2011), Laplace approximations (Yang et al., 2024), and Markov Chain Monte Carlo (MCMC) (Chen et al., 2014; Tran et al., 2022).

Within the **AdaPTS** framework, we focus in particular on variational inference (VI) for VAE adapters and on Monte Carlo dropout (Gal & Ghahramani, 2016) as an approximate form of VI for carrying out inference over $\theta$.

**Variational AutoEncoders.** Following the Bayesian perspective on adapters, VAE assume a prior distribution over the latent representation $\mathbf{Z}$, typically $\mathcal{N}(0, \mathbf{I})$. The encoder then outputs parameters of the posterior distribution $q_\phi(\mathbf{Z}|\mathbf{X})$, and the decoder generates reconstructions of predictions $\hat{\mathbf{Y}} \sim p_\theta(\mathbf{Y}|\mathbf{X}, f_{\text{FM}}(\mathbf{Z}))$ where $\theta$ parametrize a likelihood model $p$. In the context of adapters, we define

the training objective of the VAE, which brings together the forecasting loss and a regularization term, similarly to the *evidence lower bound* (ELBO) (Kingma & Welling, 2013) objective, in the following proposition.

**Proposition 4.1** (VAE adapter training objective)**.** *The training objective for the* VAE *adapter is the maximization of an* ELBO-*like lower bound on the marginal likelihood of the target* $\mathbf{Y}$:

$$\log p_\theta(\mathbf{Y}|\mathbf{X}, f_{\text{FM}}) \geq \mathbb{E}_{q_\phi(\mathbf{Z}|\mathbf{X})}\left[\log p_\theta(\mathbf{Y}|\mathbf{X}, f_{\text{FM}}(\mathbf{Z}))\right] \\ - \text{KL}\left(q_\phi(\mathbf{Z}|\mathbf{X}) \,\|\, p(\mathbf{Z})\right),$$

*where* KL *denotes the Kullback-Leibler divergence.*

The derivation of this lower bound and a discussion on the implications of each term of the loss are deferred to Appendix A.2.

*Remark* 4.2. In practice, we use the Gaussian likelihood as our likelihood model: $p_\theta(\mathbf{Y}|\mathbf{X}, f_{\text{FM}}(\mathbf{Z})) = \mathcal{N}(\mathbf{Y}; \hat{\mathbf{Y}}, \sigma^2\mathbf{I})$, with $\hat{\mathbf{Y}} = \text{dec}_\theta(f_{\text{FM}}(\mathbf{Z}))$. In this case, the forecasting loss term boils down to the MSE objective in Eq. (2) up to a multiplicative and additive noise-related constants: $\log \mathcal{N}(\mathbf{Y}; \hat{\mathbf{Y}}, \sigma^2\mathbf{I}) = -\frac{1}{2\sigma^2}\|\mathbf{Y} - \hat{\mathbf{Y}}\|_{\text{F}}^2 - \frac{HD}{2}\log(2\pi\sigma^2)$. Notice that one can also learn a model of the noise where $\text{dec}_\theta(f_{\text{FM}}(\mathbf{Z})) = [\mu_\theta(\mathbf{Y}|\mathbf{X}, f_{\text{FM}}(\mathbf{Z})), \sigma_\theta(\mathbf{Y}|\mathbf{X}, f_{\text{FM}}(\mathbf{Z}))]$.

*Remark* 4.3. The KL divergence regularization term can be multiplied by a scaling factor $\beta$ to control the disentanglement – independence of the latent representation components. This results in $\beta$-VAE (Higgins et al., 2017), that we refer to as the VAE adapter throughout the paper.

**Dropout as approximate VI.** Dropout (Srivastava et al., 2014) can be interpreted as a form of variational inference, where a variational distribution is imposed over the weights of a neural network (Gal & Ghahramani, 2016). Specifically, applying dropout during training corresponds to approximating a posterior over the weights using a Bernoulli distribution. This perspective allows the deterministic models to be transformed into probabilistic models by introducing stochasticity through dropout.

**Adapters and partially stochastic Bayesian neural networks.** Treating adapters in a Bayesian manner while keeping the FM fixed aligns with the concept of partially stochastic Bayesian neural networks, which provides theoretical guarantees on universal conditional density estimation (Sharma et al., 2023). This framework ensures that the model can approximate any conditional density, provided that stochasticity is introduced early enough in the architecture and that the number of stochastic units matches or exceeds the output dimension. Using probabilistic adapters, we comply with these conditions by making the encoder stochastic, allowing the learned latent space to capture uncertainty while leveraging the FM's fixed parameters.

---

[3]In the case of Chronos, this distribution is obtained through a categorical distribution (with *softmax* probabilities) over a tokenized space of the time series values.

| Dataset | H | No adapter | with adapter | | | | |
|---|---|---|---|---|---|---|---|
| | | Moment | PCA | LinearAE | dropoutLAE | LinearVAE | VAE |
| ETTh1 | 96 | $0.411_{\pm 0.012}$ | $0.433_{\pm 0.001}$ | $0.402_{\pm 0.002}$ | $\mathbf{0.395}_{\pm \mathbf{0.003}}$ | $0.400_{\pm 0.001}$ | $0.404_{\pm 0.001}$ |
| | 192 | $\mathbf{0.431}_{\pm \mathbf{0.001}}$ | $0.440_{\pm 0.000}$ | $0.452_{\pm 0.002}$ | $0.446_{\pm 0.001}$ | $0.448_{\pm 0.002}$ | $\mathbf{0.431}_{\pm \mathbf{0.001}}$ |
| Illness | 24 | $2.902_{\pm 0.023}$ | $2.98_{\pm 0.001}$ | $2.624_{\pm 0.035}$ | $2.76_{\pm 0.061}$ | $2.542_{\pm 0.036}$ | $\mathbf{2.461}_{\pm \mathbf{0.008}}$ |
| | 60 | $3.000_{\pm 0.004}$ | $3.079_{\pm 0.000}$ | $3.110_{\pm 0.127}$ | $2.794_{\pm 0.015}$ | $\mathbf{2.752}_{\pm \mathbf{0.040}}$ | $2.960_{\pm 0.092}$ |
| Weather | 96 | $0.177_{\pm 0.010}$ | $0.176_{\pm 0.000}$ | $0.169_{\pm 0.000}$ | $\mathbf{0.156}_{\pm \mathbf{0.001}}$ | $0.161_{\pm 0.001}$ | $0.187_{\pm 0.001}$ |
| | 192 | $0.202_{\pm 0.000}$ | $0.208_{\pm 0.001}$ | $\mathbf{0.198}_{\pm \mathbf{0.001}}$ | $0.200_{\pm 0.001}$ | $0.204_{\pm 0.000}$ | $0.226_{\pm 0.000}$ |
| ExchangeRate | 96 | $\mathbf{0.130}_{\pm \mathbf{0.011}}$ | $0.147_{\pm 0.000}$ | $0.167_{\pm 0.013}$ | $\mathbf{0.130}_{\pm \mathbf{0.011}}$ | $0.243_{\pm 0.039}$ | $0.455_{\pm 0.010}$ |
| | 192 | $\mathbf{0.210}_{\pm \mathbf{0.002}}$ | $0.222_{\pm 0.000}$ | $0.304_{\pm 0.005}$ | $0.305_{\pm 0.013}$ | $0.457_{\pm 0.020}$ | $0.607_{\pm 0.021}$ |

Table 1: Performance comparison between the baseline `Moment` model without adapters against different adapter architectures (`PCA`, `LinearAE`, `dropoutLinearAE`, `LinearVAE`, and `VAE`), for multivariate long-term forecasting with different horizons $H$. We display the average test MSE $\pm$ standard error obtained on 3 runs with different seeds. **Best** results are in bold, with lower values indicating better performance.

# 5. Experiments & Results

In this section, we empirically demonstrate quantitative and qualitative superiority of **AdaPTS** for multivariate long-term time series forecasting on common benchmarks. We show that the proposed adapter approach yields an improvement to `Moment`, a commonly used time series forecasting foundation model, over different datasets and prediction horizons. We then extend the analysis to other foundation models such as `TTM` and `TimesFM`.

## 5.1. Time series forecasting

**Datasets.** Our experiments are conducted on four publicly available real-world multivariate time series datasets, commonly used for long-term forecasting (Ilbert et al., 2024; Wu et al., 2021; Chen et al., 2023; Nie et al., 2023; Zeng et al., 2023). These datasets include the Electricity Transformer Temperature dataset (ETTh1) (Zhou et al., 2021), ExchangeRate (Lai et al., 2018), Weather (Max Planck Institute, 2021), and Influenza-like Illness (U.S. Centers for Disease Control and Prevention, 2024). All time series are segmented with an input length of $L = 512$, prediction horizons $H \in [96, 192]$ and $H \in [24, 60]$ for the Illness dataset, and a stride of 1, meaning each subsequent window is shifted by one step. These datasets (detailed in Appendix C.1) originate from various application domains, enabling a comprehensive evaluation of our framework across diverse real-world scenarios.

**Baseline.** We compare our method against the vanilla application of $\text{Moment}_s$ (where $s$ stands for *small*) from the `Moment` family of models (Goswami et al., 2024). This means that for each dataset, we apply $\text{Moment}_s$ independently to each feature. Additionally, we compare our learning-based adapters against `PCA`, an adapter that has

been used in the literature for model-based reinforcement learning (Benechehab et al., 2025) and time series classification (Feofanov et al., 2024; 2025).

**AdaPTS improves the performance of `Moment`.** We present the forecasting error measured by the Mean Squared Error (MSE) in Table 1 and the Mean Absolute Error (MAE) in Appendix D. On the ETTh1 dataset with a prediction horizon $H = 96$, all adapter-based variants outperform the baseline `Moment` model, with `dropoutLinearAE` achieving the best performance, showing an 8% improvement. Similar results are observed for the Illness dataset, where all adapters improve over the baseline. Notably, the `VAE` achieves a significant 15% improvement, reducing the MSE from 2.902 to 2.461 at $H = 24$. In the Weather dataset, the `dropoutLinearAE` adapter shows the best improvement across all adapter architectures for $H = 96$, while its deterministic counterpart, `LinearAE`, takes the lead at $H = 192$. The results on the ExchangeRate dataset are mixed, with some adapters matching the baseline performance (`dropoutLinearAE` at $H = 96$) while others show degraded performance, particularly at a longer prediction horizon ($H = 192$), which is also observed for the ETTh1 dataset. Overall, **AdaPTS** improves the forecasting accuracy of `Moment` in 5 out of the 8 considered tasks, matches its performance in 2, and degrades performance in 1 task.

**Beyond `Moment`.** To validate our approach at a broader scope, we tested **AdaPTS** against other TSFMs with distinct architectural characteristics: (1) `TTM`, an all-MLP architecture that contrasts with the transformer-based designs prevalent in other foundation models, (2) `TimesFM`, a decoder-only transformer model (in contrast to `Moment`'s encoder-only architecture). Table 3 summarizes the characteristics

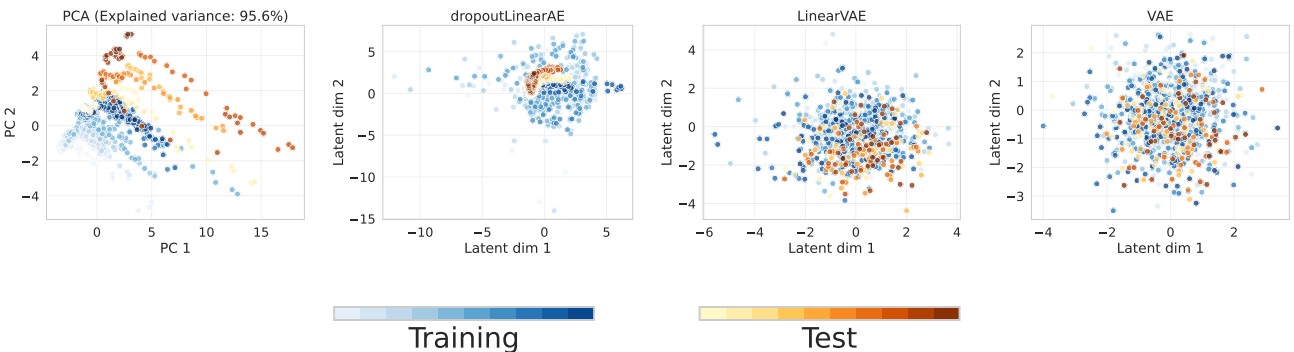

Figure 3: Visualization of the latent representation obtained by different adapters (with number of components equal to 2) on Illness($H = 24$). Shaded colors indicate the time dimension, with lighter colors representing earlier timesteps.

of these models, all of which have been integrated into our package (see Section 6). We evaluated these foundation models on the Illness($H = 24$) benchmark, measuring the relative percentage improvement achieved through optimal adapter selection and corresponding hyperparameters.

Table 2: MSE relative improvement on the Illness($H = 24$) task ($\pm$ the standard error).

| FM | Optimal adapter | MSE Imp (%) |
|---|---|---|
| TTM | VAE ($\beta = 2, \sigma = 1$) | $12.35_{\pm 2.51}$ |
| TimesFM | dropoutLAE ($p = 0.1$) | $3.64_{\pm 2.81}$ |

Table 2 reveals varying performance improvements across different foundation models, with TTM exhibiting substantial gains while TimesFM showing more modest enhancements. However, these findings should be interpreted as preliminary as evaluation across additional datasets and benchmarks is necessary to fully characterize the performance of different foundation models, a direction for ongoing research. This observation might also suggest that the effectiveness of the adapter-based approach depends on the underlying foundation model.

### 5.2. Dimensionality Reduction

Fig. 4 illustrates the impact of varying the adapter's latent space dimension $D'$ on the forecasting performance across different adapters. For the ETTh1 dataset with a 96-step horizon, all adapter architectures achieve optimal performance at 7 components (matching the original feature count), with MSE values consistently lower than the baseline. Notably, at just 5 components, all adapters (except the PCA baseline) match the baseline score, demonstrating the suitability of our framework for low-resource setups through dimensionality reduction. The Illness dataset ($H = 24$) presents more compelling results, as the VAE adapter achieves significantly optimal performance with

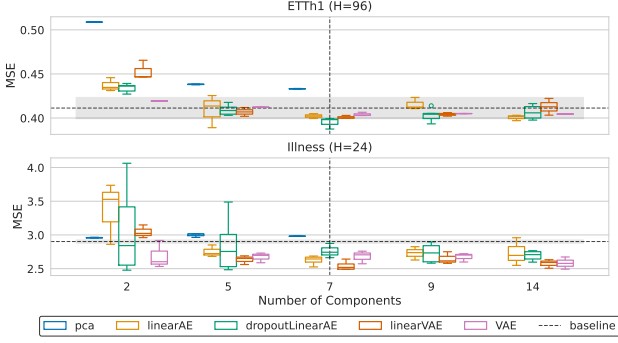

Figure 4: Impact of the number of components on the model performance. The horizontal dashed line indicates the performance of Moment without adapters. The shaded area represents its standard deviation, and the vertical line indicates the number of original features.

only 2 components, underscoring the potential of our approach for cost-effective adaptation of time series foundation models. Ultimately, we find that expanding dimensionality beyond the original feature count does not yield further improvements, as no adapter shows notable enhancements past this point.

### 5.3. Interpretability of the latent representations

Fig. 3 compares the representation learning capabilities of different adapters on the Illness ($H = 24$) dataset, focusing on their ability to distinguish between training and test data. To visualize the raw dataset, we employ PCA for dimensionality reduction, retaining only two principal components, which is justified by the 95.6% explained variance. When representing the training and test datasets in the space of the first two principal components, we observe a clear distribution shift, potentially complicating the forecasting task for the baseline foundational model. In contrast, using **AdaPTS** results in well-overlapping Gaussian distributions for the

training and test data in the latent space. This demonstrates our framework's ability to enforce a structured, isotropic representation that mitigates distribution shift. This effect is particularly pronounced with the `VAE` adapter and, to a lesser extent, with `LinearVAE` and `dropoutLinearAE`.

The findings emphasize the advantages of `VAE` in managing distribution shift, a critical challenge in time series representation learning. By modeling uncertainty and enforcing a continuous latent space, `VAE` enhance generalization, making them especially valuable for real-world applications where test distributions differ from training data. This aligns with the objective of utilizing adapters in foundational models to optimize zero-shot performance, ensuring robustness across various tasks without extensive fine-tuning.

### 5.4. On the calibration of the probabilistic adapters

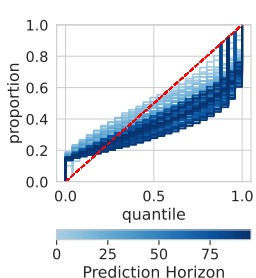

Figure 5: Reliability diagram for the first feature of the ETTh1 ($H = 96$) dataset using `LinearVAE`.

To evaluate the calibration of our adapter-based probabilistic forecasters, we use quantile calibration (Gneiting et al., 2007) as depicted in the reliability diagram in Fig. 5. In an ideal scenario, a well-calibrated probabilistic forecast should align with the red dashed diagonal, indicating that the empirical proportion of observations falls within the predicted quantiles at the expected rate. The overall conclusion is that we observe a gradual deviation from ideal calibration as the prediction horizon increases (darker shades). While early prediction horizons display reasonably well-calibrated predictions, longer-horizon forecasts systematically underestimate uncertainty, as shown by the curve falling below the diagonal. This indicates that observed values exceed predicted quantiles more frequently than expected, suggesting that the predictive distribution becomes too narrow, resulting in overconfident forecasts.

### 5.5. Ablation studies

**Influence of $\sigma$ and $\beta$ in the VAE Adapter.** Fig. 6 illustrates an ablation study examining the $\beta$ parameter in $\beta$-`VAE` and the noise scale $\sigma$ of the likelihood model applied to the prediction $\hat{\mathbf{Y}}$, assessing their effects on MSE and Expected Calibration Error (ECE) (Guo et al., 2017). The MSE heatmap (left) demonstrates that increasing $\beta$ generally diminishes MSE, with the lowest values observed at $\beta = 2.0$ and $\beta = 4.0$, particularly for higher $\log \sigma^2$. This indicates that stronger regularization through $\beta$ can enhance

forecasting accuracy, possibly due to the disentangling effect of regularization towards a prior distribution with statistically independent components. Conversely, the ECE heatmap (right) shows that higher $\beta$ and $\log \sigma^2$ values result in lower calibration error, with optimal results at $\beta = 4.0$ and $\log \sigma^2 = 3.0$. This outcome is anticipated, as larger values of $\beta$ and $\sigma$ mitigate overfitting, where the model tends to exhibit overconfidence in its predictions. Additionally, it is observed that maintaining a fixed $\sigma$ during training generally outperforms including it in the optimization loop, a configuration denoted as *auto* in Fig. 6.

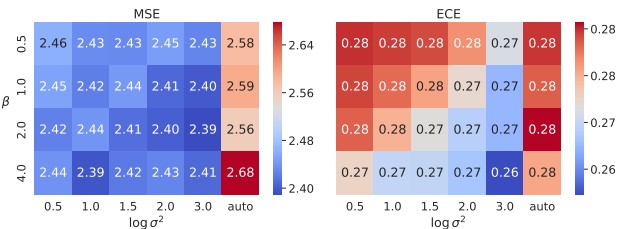

Figure 6: $\beta$ and $\log \sigma^2$ `VAE` hyperparameters ablation on the Illness($H = 24$) dataset. For reference, the `Moment` baseline score on this task is $2.902_{\pm 0.023}$.

**LinearAE components.** The ablation study presented in Fig. 7 examines the performance of different components of the linear autoencoder adapter (`LinearAE`) across three datasets: ETTh1, Weather, and ExchangeRate. The figure compares the full linear autoencoder with its encoder-only (`LinearEncoder`) and decoder-only (`LinearDecoder`) variants. Overall, the results reveal that the decoder component of the linear autoencoder plays the most important role. The encoder-only variant's contribution varies, being more impactful in the Weather dataset compared to ETTh1 and ExchangeRate. These findings highlight the significance of the decoder in the `LinearAE` adapter and suggest that, in the deterministic case, a decoder might be sufficient to capture feature dependencies.

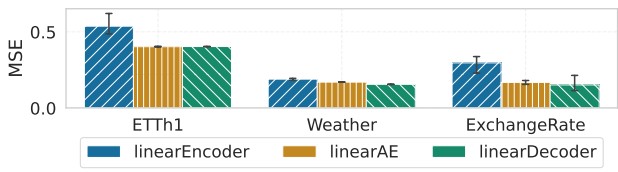

Figure 7: LinearAE components ablation.

Nevertheless, as shown in our previous experiments, particularly Table 1, probabilistic adapters generally outperformed the deterministic ones. This underscores the importance of the encoder as well, which is responsible for approximating the posterior distribution in the latent space—a mechanism inherent to our probabilistic framework.

# 6. The **AdaPTS** Python package

To facilitate reproducibility and promote widespread adoption of our framework, we have developed **AdaPTS** as a Python package that provides a unified interface for adapting time series foundation models to multivariate probabilistic forecasting tasks. The package is designed with modularity, extensibility, and ease of use in mind, enabling researchers and practitioners to seamlessly integrate our adapter-based approach into their workflows.

## 6.1. Unified Foundation Model Interface

A key feature of the **AdaPTS** package is the development of a consistent interface across different time series foundation models. The package currently supports the foundation model families listed in Table 3.

Table 3: Time Series Foundation Models supported in the `adapts` Python package. MV: Multivariate support; P: Probabilistic forecasting.

| Model | MV | P | Link (HF) |
|---|---|---|---|
| Moment | ✗ | ✗ | autonlab/moment |
| TTM | ✗ | ✗ | ibm-granite/ttm |
| TimesFM | ✗ | ✗ | google/timesfm |
| Chronos | ✗ | ✓ | amazon/chronos |
| Moirai | ! | ✓ | salesforce/moirai |

! `Moirai` uses flatten + attention bias approach to MV.

The package abstracts the complexities inherent in working with different foundation models through the *ICLTrainer* class hierarchy. Each foundation model family is implemented as a specialized child class that handles model-specific initialization, preprocessing, and inference procedures while maintaining a unified API. This architectural design enables users to effortlessly switch between different foundation models and implement custom *ICLTrainer* classes for newly-released or proprietary foundation models.

## 6.2. Training modes

Our framework supports multiple adapter training paradigms to handle different experimental scenarios:

- **Supervised Training.** This constitutes the primary training methodology analyzed throughout the paper and showcased in Fig. 1b. The adapter components (encoder and decoder) are optimized to minimize the forecasting MSE loss (Eq. (2)) while the frozen foundation model generates predictions within the learned latent space.

- **Unsupervised Training.** Adapters are trained using

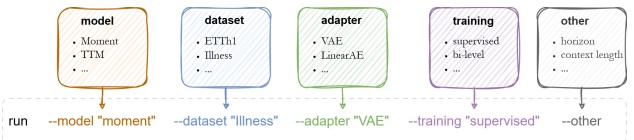

Figure 8: Command-line interface for configuring and running experiments within **AdaPTS**.

a reconstruction loss, where the encoder-decoder pair learns to reconstruct the input time series. This approach is useful for representation learning and can serve as a pretraining step before supervised fine-tuning.

- **Fine-tuning support.** This training mode permits to fine-tune the foundation model parameters alongside adapter training. This is particularly relevant for models like `Moment`, where the linear forecasting head (denoted as `LP` for linear probing in (Goswami et al., 2024)) is not pre-trained. Additionally, this can be implemented through a **bi-level** approach, where the foundation model undergoes secondary fine-tuning after completing adapter training.

Fig. 8 illustrates the command-line interface for running the **AdaPTS** approach, where users can configure their experiments by specifying the foundation model, dataset, adapter type, and training method through simple script parameters.

# 7. Conclusion

In this paper, we investigate how pre-trained univariate time series foundation models can be adapted for probabilistic multivariate forecasting. Our method, **AdaPTS**, offers a novel approach to training feature space transformations that facilitate uncertainty quantification and enhance the performance of baseline foundation models. Through a series of experiments, we demonstrate that our framework improves forecasting accuracy, provides reasonably well-calibrated uncertainty estimates, reduces inference cost through dimensionality reduction, and offers interpretable feature space latent representations.

**Limitations & Future directions.** While our primary analysis focuses on `Moment`, our approach generalizes to other foundation models, several of which have been integrated into our publicly available package. Extensive experimentation across different foundation models and benchmark datasets remains necessary to fully characterize our approach's potential. Furthermore, while we use variational inference for computational efficiency, exploring alternative methods such as MCMC could enhance uncertainty estimation and calibration, albeit with increased computational overhead.

## Impact statement

This paper presents work whose goal is to advance the field of Machine Learning. There are many potential societal consequences of our work, none which we feel must be specifically highlighted here.

## Reproducibility Statement

In order to ensure reproducibility we release the code at https://github.com/abenechehab/AdaPTS. The accompanying Python package is detailed in Section 6 while more implementation details and hyperparameters are listed in Appendix C.2.

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

# Appendix

**Outline.** In Appendix A, we provide the proofs and a discussion on Proposition 3.3 and Proposition 4.1. We then provide a perspective on Normalizing Flows as adapters in Appendix B. The experimental setup is presented in Appendix C, including all the implementation details in Appendix C.2. Finally we showcase some additional results in Appendix D.

## Table of Contents

# A. Theoretical analysis

### A.1. Proof of Proposition 3.3

We begin by restating the proposition, and its underlying assumptions:

**Assumption A.1.** $\mathbf{W}_\varphi$ has full rank: $\text{rank}(\mathbf{W}_\varphi) = D$, insuring its invertibility.

**Assumption A.2.** For ease of derivation, we consider a similar linear parametrization for the foundation model: $f_{\text{FM}}(\mathbf{X}) = \mathbf{W}_{\text{FM}}^\top \mathbf{X} + \mathbf{b}_{\text{FM}} \mathbf{1}^\top$ where $\mathbf{W}_{\text{FM}} \in \mathbb{R}^{L \times H}$, $\mathbf{b}_{\text{FM}} \in \mathbb{R}^H$, and $\mathbf{1}$ a vector of ones of dimension $D$.

**Proposition A.3** (Optimal linear adapter). *Under Assumption 3.1 and Assumption 3.2, the* closed-form *solution of the problem:*

$$\mathcal{L}(\mathbf{W}_\varphi) = \|\mathbf{Y} - (\mathbf{W}_{FM}^\top \mathbf{X} \mathbf{W}_\varphi + \mathbf{b}_{FM} \mathbf{1}^\top)\mathbf{W}_\varphi^{-1}\|_F^2 \tag{5}$$

*writes as:*

$$\mathbf{W}_\varphi^* = (\mathbf{B}^\top \mathbf{A})^+ \mathbf{B}^\top \mathbf{B} \tag{6}$$

*where* $\mathbf{W}_\varphi^* = \arg\min_{\mathbf{W}_\varphi \in \mathcal{GL}_D(\mathbb{R})} \mathcal{L}(\mathbf{W}_\varphi)$, $\mathbf{A} = \mathbf{Y} - \mathbf{W}_{FM}^\top \mathbf{X}$, $\mathbf{B} = \mathbf{b}_{FM}\mathbf{1}^\top$, *and* $(\mathbf{B}^\top\mathbf{A})^+$ *denoting the* pseudo-inverse *operator.*

*Proof.* We begin by expanding the loss function:

$$\mathcal{L}(\mathbf{W}_\varphi) = \|\mathbf{Y} - (\mathbf{W}_{\text{FM}}^\top \mathbf{X} \mathbf{W}_\varphi + \mathbf{b}_{\text{FM}} \mathbf{1}^\top)\mathbf{W}_\varphi^{-1}\|_F^2$$
$$= \|\mathbf{A} - \mathbf{B}\mathbf{W}_\varphi^{-1}\|_F^2$$

where $\mathbf{A} = \mathbf{Y} - \mathbf{W}_{FM}^\top \mathbf{X}$ and $\mathbf{B} = \mathbf{b}_{FM}\mathbf{1}^\top$. Expanding the Frobenius norm:

$$\mathcal{L}(\mathbf{W}_\varphi) = \text{Tr}\left((\mathbf{A} - \mathbf{B}\mathbf{W}_\varphi^{-1})^\top(\mathbf{A} - \mathbf{B}\mathbf{W}_\varphi^{-1})\right)$$

Taking the gradient with respect to $\mathbf{W}_\varphi^{-1}$ yields:

$$\frac{\partial \mathcal{L}}{\partial \mathbf{W}_\varphi^{-1}} = -2\mathbf{B}^\top \mathbf{A} + 2\mathbf{B}^\top \mathbf{B} \mathbf{W}_\varphi^{-1}$$

Knowing that $\mathbf{W}_\varphi$ is invertible, We have that: $\frac{\partial \mathcal{L}}{\partial \mathbf{W}_\varphi} = -\mathbf{W}_\varphi^{-\top} \frac{\partial \mathcal{L}}{\partial \mathbf{W}_\varphi^{-1}} \mathbf{W}_\varphi^{-\top}$

hence

$$\frac{\partial \mathcal{L}}{\partial \mathbf{W}_\varphi} = -2\mathbf{W}_\varphi^{-T}\left(\mathbf{B}^\top \mathbf{A} - \mathbf{B}^\top \mathbf{B} \mathbf{W}_\varphi^{-1}\right)\mathbf{W}_\varphi^{-T}.$$

Setting $\frac{\partial \mathcal{L}}{\partial \mathbf{W}_\varphi} = 0$ and multiplying both sides by $\mathbf{W}_\varphi^\top$, we obtain:

$$\mathbf{B}^\top \mathbf{A} = \mathbf{B}^\top \mathbf{B} \mathbf{W}_\varphi^{-1}.$$

Multiplying both sides by $\mathbf{W}_\varphi$:

$$\mathbf{B}^\top \mathbf{A} \mathbf{W}_\varphi = \mathbf{B}^\top \mathbf{B}.$$

Finally applying the pseudo-inverse to solve for $\mathbf{W}_\varphi$ gives our final result:

$$\mathbf{W}_\varphi^* = (\mathbf{B}^\top \mathbf{A})^+ \mathbf{B}^\top \mathbf{B}.$$

Given the convexity of $\mathcal{L}(\mathbf{W}_\varphi)$ (which follows from the convexity of the Frobenius norm $\|\cdot\|_F^2$, the inverse operation, and an affine transformation), we conclude that $\mathbf{W}_\varphi^*$ is a global solution for Eq. (3).

*Remark* A.4. We make use of the pseudo-inverse due to the current construction of the matrix $\mathbf{B}$ (with identical rows) which implies that the product $\mathbf{B}^\top \mathbf{A}$ is degenerate. To bypass this limitation and further ensure the invertibility of $\mathbf{W}_\varphi^*$, we can revisit the definition of the foundation model in Assumption 3.2 to include channel dependent biases and ensure a full rank matrix $\mathbf{B}$.

$\square$

### A.2. Proof of Proposition 4.1

To derive the evidence lower bound (ELBO) used in the training objective of the VAE adapter, we start from the marginal likelihood of the observed data $\mathbf{Y}$ given the inputs $\mathbf{X}$ and foundation model $f_{\mathrm{FM}}$. The marginal likelihood is expressed as:

$$\log p_\theta(\mathbf{Y}|\mathbf{X}, f_{\mathrm{FM}}) = \log \int p_\theta(\mathbf{Y}|\mathbf{X}, f_{\mathrm{FM}}(\mathbf{Z}))p(\mathbf{Z})\, d\mathbf{Z}, \tag{7}$$

where $\mathbf{Z}$ is the latent variable, $p_\theta(\mathbf{Y}|\mathbf{X}, f_{\mathrm{FM}}(\mathbf{Z}))$ is the likelihood model parameterized by $\theta$, and $p(\mathbf{Z})$ is the prior distribution over the latent variable $\mathbf{Z}$.

Direct optimization of this marginal likelihood is generally intractable due to the integration over $\mathbf{Z}$. To make this optimization feasible, we introduce a variational distribution $q_\phi(\mathbf{Z}|\mathbf{X})$, parameterized by $\phi$, as an approximation to the true posterior $p_\theta(\mathbf{Z}|\mathbf{X}, \mathbf{Y}, f_{\mathrm{FM}})$. Using $q_\phi(\mathbf{Z}|\mathbf{X})$, we can reformulate the log-marginal likelihood as follows:

$$\log p_\theta(\mathbf{Y}|\mathbf{X}, f_{\mathrm{FM}}) = \log \int q_\phi(\mathbf{Z}|\mathbf{X}) \frac{p_\theta(\mathbf{Y}|\mathbf{X}, f_{\mathrm{FM}}(\mathbf{Z}))p(\mathbf{Z})}{q_\phi(\mathbf{Z}|\mathbf{X})}\, d\mathbf{Z} \tag{8}$$

$$= \log \mathbb{E}_{q_\phi(\mathbf{Z}|\mathbf{X})}\left[\frac{p_\theta(\mathbf{Y}|\mathbf{X}, f_{\mathrm{FM}}(\mathbf{Z}))p(\mathbf{Z})}{q_\phi(\mathbf{Z}|\mathbf{X})}\right]. \tag{9}$$

Using Jensen's inequality, we can derive a lower bound on this log-marginal likelihood:

$$\log p_\theta(\mathbf{Y}|\mathbf{X}, f_{\mathrm{FM}}) \geq \mathbb{E}_{q_\phi(\mathbf{Z}|\mathbf{X})}\left[\log \frac{p_\theta(\mathbf{Y}|\mathbf{X}, f_{\mathrm{FM}}(\mathbf{Z}))p(\mathbf{Z})}{q_\phi(\mathbf{Z}|\mathbf{X})}\right] \tag{10}$$

$$= \mathbb{E}_{q_\phi(\mathbf{Z}|\mathbf{X})}\left[\log p_\theta(\mathbf{Y}|\mathbf{X}, f_{\mathrm{FM}}(\mathbf{Z}))\right] - \mathbb{E}_{q_\phi(\mathbf{Z}|\mathbf{X})}\left[\log \frac{q_\phi(\mathbf{Z}|\mathbf{X})}{p(\mathbf{Z})}\right]. \tag{11}$$

The second term can be rewritten as the Kullback-Leibler (KL) divergence between the variational posterior $q_\phi(\mathbf{Z}|\mathbf{X})$ and the prior $p(\mathbf{Z})$:

$$\mathrm{KL}\left(q_\phi(\mathbf{Z}|\mathbf{X}) \,\|\, p(\mathbf{Z})\right) = \mathbb{E}_{q_\phi(\mathbf{Z}|\mathbf{X})}\left[\log \frac{q_\phi(\mathbf{Z}|\mathbf{X})}{p(\mathbf{Z})}\right]. \tag{12}$$

Substituting this into the inequality, we obtain the evidence lower bound (ELBO):

$$\log p_\theta(\mathbf{Y}|\mathbf{X}, f_{\mathrm{FM}}) \geq \mathbb{E}_{q_\phi(\mathbf{Z}|\mathbf{X})}\left[\log p_\theta(\mathbf{Y}|\mathbf{X}, f_{\mathrm{FM}}(\mathbf{Z}))\right] - \mathrm{KL}\left(q_\phi(\mathbf{Z}|\mathbf{X}) \,\|\, p(\mathbf{Z})\right). \tag{13}$$

The ELBO consists of two terms:

- The *forecasting* term, $\mathbb{E}_{q_\phi(\mathbf{Z}|\mathbf{X})}\left[\log p_\theta(\mathbf{Y}|\mathbf{X}, f_{\mathrm{FM}}(\mathbf{Z}))\right]$, which measures how well the model can reconstruct $\mathbf{Y}$ given the latent variable $\mathbf{Z}$.

- The *regularization* term, $\mathrm{KL}\left(q_\phi(\mathbf{Z}|\mathbf{X}) \,\|\, p(\mathbf{Z})\right)$, which encourages the variational posterior to stay close to the prior distribution $p(\mathbf{Z})$.

Thus, maximizing the ELBO provides a tractable way to train the parameters $\theta$ and $\phi$ by optimizing the balance between forecasting accuracy and latent space regularization. $\square$

## B. Normalizing Flows

Normalizing Flows make use of invertible transformations to map a simple base distribution (e.g. Gaussian) to a complex data distribution. Each transformation $T$ is designed to maintain invertibility and efficient Jacobian computation. The transformation is applied iteratively: $\mathbf{Z} = T_k \circ T_{k-1} \circ \cdots \circ T_1(\mathbf{X})$. Current Normalizing Flow instantiations (e.g. `RealNVP`) make use of generic invertible transformations such as *coupling flows*; the latters can be parametrized using a neural network leading to powerful non-linear generative models that are trained to maximize the data log-likelihood:

$$\log p(\mathbf{X}) = \log p(\mathbf{Z}) + \sum_{i=1}^{k} \log \left| \det \frac{\partial T_i(\cdot; \theta)}{\partial \mathbf{Z}_{i-1}} \right|$$

where $\theta$ denote the parameters of the non-linear parametrization of the invertible transformations $T_i$, and $\mathbf{Z}_{i-1}$ is the output of the transformation $T_{i-1}$. In the context of time series adapters, we directly optimize the parameters of the transformations based on their direct and inverse application on the time series forecasting problem:

$$\mathcal{L}_{\text{flow}} = \|\mathbf{Y} - T_1^{-1} \circ T_2^{-1} \circ \cdots \circ T_k^{-1}(\\ f_{\text{FM}}(T_k \circ T_{k-1} \circ \cdots \circ T_1(\mathbf{X}; \theta)); \theta)\|_F^2$$

where the encoder is represented by the series of direct transformations: $\text{enc}(\cdot) = T_k \circ T_{k-1} \circ \cdots \circ T_1(\cdot; \theta)$, and respectively the decoder by the series of inverse transformations $\text{dec}(\cdot) = T_1^{-1} \circ T_2^{-1} \circ \cdots \circ T_k^{-1}(\cdot; \theta)$.

As defined here, Normalizing Flows suffer from the constraint of keeping the same dimension in both original and learned representation space. For this purpose, we investigate coupling a normalizing flow with a linear encoder-decoder type of architecture to enable dimensionality reduction prior to applying the transformation $T_i$. The parameters of the additional encoder and decoder are then jointly trained to optimize the learning objective $\mathcal{L}_{\text{flow}}$.

Given that the parameters of the encoder and the decoder are shared in Normalizing Flows, the gradient-based optimization within our framework receives conflicting directions due to gradient flow from both the direct and inverse transformations simultaneously. We discovered that this adapter construction was challenging to optimize in practice, and we defer the exploration of this direction to future research endeavors.

## C. Experimental setup

### C.1. Datasets

Table 4: Characteristics of the multivariate time series datasets used in our experiments with various sizes and dimensions.

| Dataset | ETTh1 | Illness | ExchangeRate | Weather |
|---|---|---|---|---|
| # features | 7 | 7 | 8 | 21 |
| # time steps | 13603 | 169 | 6791 | 51899 |
| Granularity | 1 hour | 1 week | 1 day | 10 minutes |
| (Train, Val, Test) | (8033, 2785, 2785) | (69, 2, 98) | (4704, 665, 1422) | (36280, 5175, 10444) |

### C.2. Implementation details

In this section, we describe the full **AdaPTS** framework, starting from the data preprocessing, the training algorithm, and the hyperparameters optimization.

**Preprocessing.** Given that the adapter as defined in Definition 2.1 is a feature space transformation, we start by rescaling (*StandardScaler* and *MinMaxScaler*) the data where all the timesteps are regarded as data points. To account for the temporal specificities in each batch, we use Reversible Instance Normalization (RevIn) (Kim et al., 2022) that has been proven to mitigate time-related distribution shifts in time series problems. Finally, and following the observation that `PCA` when composed with a linear adapter showed the best result in the case of correlated data (Fig. 2), we include the possibility of applying *full-component* `PCA` as part of our data pre-processing pipeline.

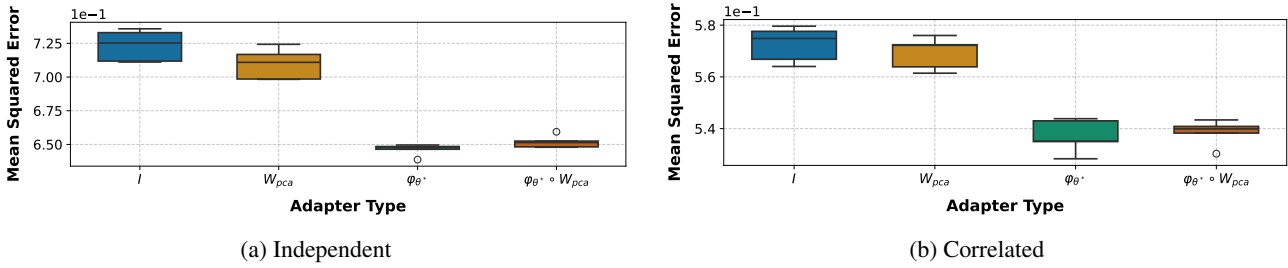

(a) Independent

(b) Correlated

Figure 9: `Moment` on simulated independent data.

**Training parameters.** After the pre-processing phase, we proceed to split the data into a *train-validation-test* sets, where the validation set serves as a tool to select the best hyperparameters for the adapter. The resulting adapter that is instantiated with the optimal hyperparameters is then tested against the unseen test dataset. For all of our experiments, we first train the linear forecasting head of `Moment` (referred to as *Linear Probing* in (Goswami et al., 2024)) with the Adam optimizer (Kingma & Ba, 2017), a batch size of 32, a one cycle scheduler starting with 0.001 as learning rate. Once the forecasting linear head is trained, we freeze its parameters and proceed to training the adapter. This is done using the Adam optimizer, a batch size of 32, a reduce on plateau scheduler starting with 0.001 as learning rate.

**Hyperparameter optimization.** In order to select the best hyperparameters for the adapter architecture we use *Ray tune* (Liaw et al., 2018) with the Heteroscedastic and Evolutionary Bayesian Optimisation solver (HEBO) (Cowen-Rivers et al., 2022) engine, reporting the average mean squred error (MSE) from *k-fold* cross validation. Table 5 shows the default hyperparameters for each considered adapter.

Table 5: Adapters hyperparameters.

| adapter | LinearAE | DropoutLinearAE | LinearVAE | VAE |
|---|---|---|---|---|
| p dropout | — | 0.1 | — | — |
| Number of layers | — | — | — | 2 |
| Hidden dimension | — | — | — | 128 |
| $\beta$ | — | — | 0.5 | 0.5 |
| $\sigma$ | — | — | 1.0 | 1.0 |

## D. Additional results

### D.1. `Moment` applied to synthetic data.

To validate the adapter *optimality* condition with large non-linear foundation models, we use `Moment` (Goswami et al., 2024). The optimal linear adapter in this case minimizes the following intractable objective:

$$\mathcal{L}(\mathbf{W}_\varphi) = \|\mathbf{Y} - f_{\text{Moment}}(\mathbf{X}\mathbf{W}_\varphi)\mathbf{W}_\varphi^{-1}\|_{\text{F}}^2 \tag{14}$$

To approximately solve this optimization problem, we instantiate $\mathbf{W}_\varphi$ as a single-linear-layer encoder denoted $\text{enc}_\theta$, and respectively the inverse transformation $\mathbf{W}_\varphi^{-1}$ as a single-linear-layer decoder denoted $\text{dec}_\theta$. We then use gradient-based optimization of the parameters $\theta$ using the Adam optimizer, aiming at solving the following optimization problem:

$$\theta^* = \arg\min_\theta \|\mathbf{Y} - \text{dec}_\theta(f_{\text{Moment}}(\text{enc}_\theta(\mathbf{X})))\|_{\text{F}}^2 \tag{15}$$

Fig. 9 shows the performance gain obtained by optimizing a linear adapter on *Moment-small* foundation model. Unlike the tractable case, we observe that in both data modalities (independent and correlated data), `PCA` has little to no improvement over the identity baseline, while $\varphi_{\theta^*}$ reaches an order of magnitude better solution. This confirms our intuition about the existence of a better solution than the identity matrix, even in the case of real-world complex foundation models.

## D.2. Mean Absolute Error

| Dataset | H | No adapter | with adapter | | | | |
|---------|---|------------|------|------|------|------|------|
| | | Moment$_{\text{small}}$ | pca | linear | dropout | linear VAE | VAE |
| ETTh1 | 96 | $0.422_{\pm 0.006}$ | $0.440_{\pm 0.000}$ | $0.423_{\pm 0.003}$ | $\mathbf{0.415_{\pm 0.002}}$ | $0.420_{\pm 0.001}$ | $0.426_{\pm 0.001}$ |
| | 192 | $\mathbf{0.436_{\pm 0.000}}$ | $0.445_{\pm 0.000}$ | $0.449_{\pm 0.003}$ | $0.450_{\pm 0.001}$ | $0.451_{\pm 0.001}$ | $0.444_{\pm 0.001}$ |
| Illness | 24 | $1.143_{\pm 0.007}$ | $1.163_{\pm 0.001}$ | $2.624_{\pm 0.035}$ | $1.156_{\pm 0.016}$ | $1.074_{\pm 0.011}$ | $\mathbf{1.057_{\pm 0.012}}$ |
| | 60 | $1.149_{\pm 0.001}$ | $1.161_{\pm 0.001}$ | $1.227_{\pm 0.030}$ | $1.173_{\pm 0.015}$ | $1.112_{\pm 0.021}$ | $\mathbf{1.105_{\pm 0.021}}$ |
| Weather | 96 | $0.232_{\pm 0.010}$ | $0.235_{\pm 0.000}$ | $0.226_{\pm 0.000}$ | $0.212_{\pm 0.001}$ | $\mathbf{0.218_{\pm 0.001}}$ | $0.243_{\pm 0.001}$ |
| | 192 | $\mathbf{0.251_{\pm 0.001}}$ | $0.260_{\pm 0.001}$ | $\mathbf{0.251_{\pm 0.001}}$ | $\mathbf{0.251_{\pm 0.000}}$ | $0.255_{\pm 0.000}$ | $0.274_{\pm 0.000}$ |
| ExchangeRate | 96 | $\mathbf{0.252_{\pm 0.010}}$ | $0.264_{\pm 0.000}$ | $0.308_{\pm 0.010}$ | $0.269_{\pm 0.012}$ | $0.376_{\pm 0.031}$ | $0.488_{\pm 0.003}$ |
| | 192 | $\mathbf{0.329_{\pm 0.001}}$ | $0.335_{\pm 0.000}$ | $0.415_{\pm 0.002}$ | $0.419_{\pm 0.010}$ | $0.513_{\pm 0.010}$ | $0.585_{\pm 0.008}$ |

Table 6: Performance comparison between the baseline `Moment` model without adapters against different adapter architectures (`PCA`, `LinearAE`, `dropoutLAE`, `LinearVAE`, and `VAE`), for multivariate long-term forecasting with different horizons $H$. We display the average test MAE $\pm$ standard error obtained on 3 runs with different seeds. **Best** results are in bold, with lower values indicating better performance.

