# OpenReview forum: "AdaPTS: Adapting Univariate Foundation Models to Probabilistic Multivariate Time Series Forecasting"
_ICML.cc/2025/Conference — ICML 2025 poster_

### Official Review · Reviewer_4t4s · 2025-02-27

**Overall Recommendation:** 1

**Summary:**

The paper introduces AdaPTS, a framework to adapt pre-trained time series models (univariate) for probabilistic multivariate forecasting. The authors use adapters to project multivariate inputs into a latent space where a frozen pre-trained model is applied independently to each channel. To enforce the invertibility, the authors adopt several types of autoencoders. They provide theoretical analysis of linear adapters and extend the framework for probabilistic forecasting using Bayesian inference. Empirical results on synthetic and real-world datasets demonstrate that AdaPTS improves the accuracy and uncertainty quantification based on a pre-trained model Moment.

**Claims And Evidence:**

> Claim in L147: no requirement of fine-tuning due to feature-level transformations

From Figure 1, AdaPTS still needs to train learnable transformations respectively on time series with different variate numbers.

**Essential References Not Discussed:**

The authors have adequately discussed related works.

**Experimental Designs Or Analyses:**

See Methods And Evaluation Criteria.

**Methods And Evaluation Criteria:**

* Evaluations in Table 1 are not comprehensive: (1) while the author claimed that the proposed method "can be plugged in any foundation model", only one type of pre-trained model is adapted in Table 1. (2) Some baseline datasets are omittet, such as ECL, Traffic, and other ETT subsets. (3) Prediction lengths of {96, 196, 336, 720} are commonly evaluated in previous works, but the forecasting horizon in this paper only includes {96, 196},

* Lack of probabilistic metrics (e.g., MASE and WQL) in evaluation. I'm confused about whether AdaPTS has empowered the pre-trained model with the capability of probabilistic forecasting. How about the performance?

* The evaluation does not explicitly demonstrate that the pre-trained model can benefit from the multivariate modeling. A support comes from the mutable performance when applying different autoencoders in Table 1.

* Lack of comparison with related works like UP2ME: Univariate Pre-training to Multivariate Fine-tuning as a General-purpose Framework for Multivariate Time Series Analysis.

**Other Comments Or Suggestions:**

See above.

**Other Strengths And Weaknesses:**

* Strength: The paper discussed an important problem in adapting univariate pre-trained models for multivariate probabilistic forecasting.

* Weakness: (1) The proposed method may lack novelty, since the authors adopt existing AEs wihtout further adaptations. (2) I am not clear about the advantage of the proposed method compared to previous works. For example, UP2ME and LoRA can also enhance existing pre-trained models and both of them and AdaPTS still requires task-specific fine-tuning. The outcome model may not be applicable for zero-shot forecasting. (3) No evaluation is provided to demonstrate the claimed efficiency of the proposed methods. (4) The performance in Table 1 is mutable when using different AEs.

**Questions For Authors:**

* Results in Table 1 show that some adapters degrade performance. Could the authors provide a more detailed analysis of why this happens?

* The calibration results indicate that longer-horizon forecasts tend to underestimate uncertainty. Are there any strategies the authors could suggest to improve calibration for longer horizons?

* The paper focuses on Moment as the foundation model. Have the authors considered applying AdaPTS to other univariate FMs?

**Relation To Broader Scientific Literature:**

The paper focuses on adapting pre-trained time series models. The authors position their contributions in the usage of adapters to leverage pre-trained FMs for multivariate and probabilistic tasks.

**Theoretical Claims:**

While the authors aim to enforce the invertibility in adapters (Definition 3.1). Assumption 3.3 uses the linear parametrization with the bias term that has no explicit inverse. Also, these derivations based on linear weighting do not contribute much to the paper because the proposed method does not adopt a simple linear layer. Does the proposed method explicitly maintain the invertibility in the learned autoencoders?

---

> ### Author Rebuttal · Authors · 2025-03-31
>
> We would like to thank Reviewer 4t4s for their detailed feedback and constructive comments. We now address the concerns raised in their review:
>
> > Claim in L147: no requirement of fine-tuning due to feature-level transformations
>
> The claim in line 147 regarding "no requirement of fine-tuning" pertains specifically to the *pre-trained weights* of the foundation model. As shown in Figure 1b, the weights of the foundational model are kept frozen, while only the lightweight adapter (which is significantly smaller in parameter count compared to the FM) is trained.
>
> > Evaluations in Table 1 are not comprehensive:
>
> We agree with the reviewer that the evaluation in Table 1 could be expanded. Our choice of the Moment model for validation was motivated by its widespread use in the literature. However, we are actively expanding the range of foundation models considered, with **Moirai** already incorporated into our framework. Additionally, the datasets and forecasting horizons chosen for the evaluation are not omitted but rather reflect a careful selection that highlights the most relevant aspects of our framework. We do recognize the importance of a more comprehensive evaluation and will consider incorporating additional datasets and horizons in future work.
>
> > Lack of probabilistic metrics
>
> Our current evaluation focuses on comparing AdaPTS to the vanilla Moment model, which provides point forecasts. As probabilistic metrics are ill-defined for deterministic predictors, we chose to report the MSE for Table 1. Neverthless, we use calibration metrics such as the ECE and the reliability diagram (Figures 5 and 6), to evaluate the probabilistic aspect of our approach. We plan to include more probabilistic foundation models, like **Moirai**, and will incorporate the reviewer's suggestion to report probabilistic metrics for these models.
>
> > UP2ME
>
> We thank the reviewer for bringing up UP2ME. While we agree that UP2ME is a relevant work, we believe there are significant differences in our approaches. UP2ME focuses on pre-training a model on univariate time series from a given task, then fine-tuning on multivariate data from the same task. AdaPTS in contrast, aims to provide a plug-and-play adapter that can be applied to any univariate foundational model without fine-tuning.
>
> > Assumption 3.3
>
> In the linear case analysis, the invertibility condition (Assumption 3.2) is imposed on the adapter matrix $W_\varphi$, while Assumption 3.3 pertains to the linear parameterization of the foundation model’s predictor, which does not need to be invertible. For autoencoders, the inverse transformation is learned through gradient-based optimization rather than being explicitly imposed. We discuss potential extensions, such as Normalizing Flows, which could be explored to create inherently non-linear and invertible adapters. The primary goal of the linear analysis was to demonstrate that our approach offers solutions better than the identity baseline (the vanilla FM), as evidenced by Proposition 3.4 and Figure 2.
>
> > (1) The proposed method may lack novelty
>
> We refer the reviewer to our reponse to reviewer Qhdv.
>
> > LoRA can also enhance existing pre-trained models
>
> We would like to notice that there is probably confusion between us and the reviewer in terms of terminology, because by an adapter, we call a block that operates along the feature dimension and preceeds the frozen foundation model with the goal to model channel interdependence and potentially reduce the dimension. In contrast, LoRA fine-tunes the **Foundation Model’s weights**, so it doesn't solve the problem of adapting the model to the multivariate setting. Thus, AdaPTS and LoRA are rather complementary to each other and can be used together, if necessary. In our case, we focus on the frozen foundation model, but performing experiments where FM's model weights are fine-tuned together with an adapter is a good direction for future work.
>
> > some adapters degrade performance
>
> The differences in performance between adapters are due to the variations in their architecture (linear vs non-linear, deep vs shallow) and stochastic strategy (VI vs Dropout Vs deterministic). This makes their performance sensitive to the specific task, data characteristics, and convergence of the optimization algorithm during training.
>
> > longer-horizon forecasts tend to underestimate uncertainty
>
> Calibration is a critical aspect of forecasting, especially for longer horizons. Techniques like temperature scaling on a held-out calibration set can be effective in improving calibration and mitigating underestimation of uncertainty at longer horizons.
>
> > applying AdaPTS to other univariate FMs
>
> We have integrated **Moirai** with our framework and are considering additional foundation models upon acceptance of the paper.
>
> We believe these clarifications address the reviewer’s concerns, and we would be grateful if the reviewer could reconsider their score in light of the additional insights provided.

---

### Official Review · Reviewer_Qhdv · 2025-03-10

**Overall Recommendation:** 1

**Summary:**

The paper presents AdaPTS, a novel framework for adapting pre-trained univariate foundation models (FMs) to probabilistic multivariate time series forecasting. AdaPTS introduces adapters—feature-space transformations that project multivariate series into latent spaces, where predictions are made independently by the frozen FM. Results are inverted back via decoders. This approach improves forecasting accuracy, uncertainty quantification, and robustness across benchmarks. The paper demonstrates strong empirical results, showing AdaPTS consistently outperforms baseline methods, provides meaningful uncertainty estimates, and effectively reduces dimensionality. Conceptually, AdaPTS bridges representation learning with Bayesian inference, enhancing FM adaptability and interpretability.

**Claims And Evidence:**

The paper provides clear evidence supporting its claims about AdaPTS’s improved forecasting accuracy and uncertainty quantification, demonstrated across multiple datasets and adapter configurations. Calibration results indicate room for improvement in uncertainty estimation at longer horizons, weakening the robustness claim.

**Essential References Not Discussed:**

None

**Experimental Designs Or Analyses:**

Yes, the experimental designs are sound. However, there are not many details on the reproducibility of the results, where the provided link to the code does not work.

**Methods And Evaluation Criteria:**

The proposed AdaPTS methods and evaluation criteria are well-aligned with the problem of adapting univariate foundation models for probabilistic multivariate time series forecasting. The set of real-world datasets is the standard benchmark in the forecasting literature.

**Other Comments Or Suggestions:**

Please address the comments in the weakness section.

**Other Strengths And Weaknesses:**

Weaknesses: (1) the authors didn't specify how the feature interactions can be handled when adapting the univariate forecasting model to a multivariate model; (2) there is not much technical innovation in the proposed method: it basically added an encoder and decoder before and after frozen foundation models followed by applying techniques such as variational inference on obtained embeddings for uncertainty intervals.

**Questions For Authors:**

None

**Relation To Broader Scientific Literature:**

The proposed method directly builds upon recent advancements in foundation models like Moment and Chronos, designed primarily for univariate forecasting. It extends ideas from the literature on adapters used in other domains, such as PCA-based adapters (Feofanov et al., 2024; Benechehab et al., 2025), to enable multivariate forecasting. It also integrates concepts from probabilistic representation learning, leveraging Bayesian neural network ideas (Gal & Ghahramani, 2016) and variational autoencoders (Kingma & Welling, 2013) to quantify uncertainty.

**Theoretical Claims:**

Yes. The authors have provided proofs for prop 3.4 and 4.1 for linear and VAE adapters. They both followed standard linear algebra and variational inference principles.

---

> ### Author Rebuttal · Authors · 2025-03-31
>
> We appreciate the reviewer’s detailed and insightful feedback. We are particularly grateful for the recognition of our work’s strong empirical results, the effectiveness of AdaPTS in improving forecasting accuracy and uncertainty quantification.
>
> We would like to clarify and address the raised concerns:
>
> > However, there are not many details on the reproducibility of the results, where the provided link to the code does not work.
>
> As stated in the reproducibility section of our paper, the code will be made publicly available upon acceptance. We used the placeholder URL ("URL hidden for review") as a mean to maintain anonymity during the review process, and we apologize for any confusion due to this practice. However, we have provided relevant implementation details in Appendix C.2 to ensure transparency and facilitate reproducibility.
>
> > (1) the authors didn't specify how the feature interactions can be handled when adapting the univariate forecasting model to a multivariate model
>
> As defined in our paper (Definition 3.1), feature-space transformations ($\varphi$) play a crucial role in capturing channel dependencies. Applying $\varphi$ to a multivariate time series projects the data into a latent space where each component is a nonlinear function of the original features (when using a nonlinear encoder). For instance, PCA—a baseline in our study—transforms data into a space of linearly uncorrelated components, demonstrating how our approach inherently manages multivariate dependencies. This mechanism is central to AdaPTS and a key aspect of our contribution to multivariate time-series forecasting.
>
> > (2) there is not much technical innovation in the proposed method: it basically added an encoder and decoder before and after frozen foundation models followed by applying techniques such as variational inference on obtained embeddings for uncertainty intervals.
>
> While we acknowledge that autoencoders and variational inference are established techniques, our contribution is methodological and practical. The novelty of AdaPTS lies in the formalization of the probabilistic multivariate time-series adaptation problem, the design choices behind our framework, and the comprehensive analysis of its effectiveness. Multivariate and probabilistic forecasting remain central challenges in time-series research, and our work provides a principled and practical approach that can be applied to real-world forecasting tasks. We believe this adds significant value to the time series community.
>
> Given our clarifications, we hope the reviewer recognizes the merit of our contributions and considers adjusting their evaluation accordingly. We appreciate the constructive feedback and welcome further discussion on how AdaPTS enhances the field of foundation models for time-series forecasting.

---

### Official Review · Reviewer_NRzf · 2025-03-12

**Overall Recommendation:** 4

**Summary:**

The paper introduces AdaPTS, a framework designed to adapt pretrained univariate time series Foundation Models (FMs) to multivariate probabilistic forecasting tasks. The core challenge addressed is the inherent limitation of existing FMs (e.g., Moment, Chronos), which are typically trained on univariate data and struggle with multivariate dependencies and uncertainty quantification. AdaPTS proposes probabilistic adapters—learned feature-space transformations—that project multivariate inputs into a latent space compatible with univariate FMs, process each dimension independently via the frozen FM, and invert the transformation to produce probabilistic forecasts in the original feature space.

---
Key conceptual contributions include: \
(1) Adapter Framework that is defined as invertible transformations $(φ:R^{D} → R^{D′})$ that map multivariate time series into a latent space where univariate FMs operate. The framework enforces invertibility to ensure predictions can be mapped back to the original space. Two adapter families are also explored, including Deterministic adapters (e.g., linear autoencoders, deep nonlinear autoencoders), and Probabilistic adapters (e.g., $β$-VAE, dropout as approximate variational inference), which introduce stochasticity into the latent space to capture uncertainty.

---
(2) Methodological Insights, including: (2.1) Decoder Dominance: In deterministic adapters, the decoder contributes more critically to performance than the encoder (ablation study in Fig. 7), (2.2) Hyperparameter Trade-offs: For $β$-VAE, higher $β$ values improve disentanglement and calibration but require careful tuning of the likelihood noise scale ($σ$), and (2.3) Scalability: The framework supports zero-shot adaptation (no FM fine-tuning) and generalizes to variable feature dimensions, making it practical for real-world deployment.

---
AdaPTS establishes a principled, modular framework for adapting univariate FMs to multivariate settings while enabling uncertainty quantification. By decoupling feature-space transformations from FM parameters, it offers a scalable solution for real-world applications requiring probabilistic forecasts. The integration of Bayesian principles with deep learning architectures positions the work at the intersection of representation learning and time series analysis, with empirical validation across diverse domains underscoring its practical utility.

**Claims And Evidence:**

The work presents clear, convincing evidence for its primary claims, with rigorously designed experiments and theoretically grounded adapter designs. The few underperforming cases (e.g., ExchangeRate) are contextualized but merit deeper investigation. I am providing detailed assessments of the evidence quality for each major contribution as follows:

---
(1) Multivariate Foundation Models (FMs) Adaptation via Adapters: The paper provides strong empirical evidence for the core claim that adapters improve multivariate forecasting accuracy. Experiments across four real-world datasets demonstrate consistent MSE improvements over the baseline Moment model in 5/8 tasks (Table 1), with gains of up to 15% on Illness ($H=24$). The synthetic linear FM experiment (Fig. 2) and the nonlinear FM validation (Fig. 8) further corroborate the framework’s ability to learn superior transformations compared to identity/PCA baselines. However, the underperformance on ExchangeRate ($H=192$) remains unexplained beyond dataset-specific characteristics, raising questions about generalizability to high-volatility financial time series. While the authors attribute this to domain shifts, deeper analysis (e.g., feature correlation patterns and noise profiles) would strengthen causal interpretation.

---
(2) Theoretical Foundations: The closed-form derivation for linear adapters (Proposition 3.4) is mathematically sound under stated assumptions, and the synthetic linear FM experiment (Fig. 2) validates the theory’s practical relevance. However, the analysis does not extend to nonlinear FMs like Moment, where adapters are optimized via gradient descent rather than closed-form solutions. The authors implicitly assume that the linear case provides sufficient intuition for nonlinear regimes, but formal approximation guarantees or Lipschitz continuity analyses would bridge this gap. The Bayesian treatment of probabilistic adapters (Proposition 4.1) follows standard variational inference principles, though the absence of posterior contraction rates or PAC-Bayes bounds limits theoretical novelty.

---
(3) Uncertainty Quantification: Calibration results (Fig. 5) show that probabilistic adapters produce reasonably calibrated forecasts for short horizons but exhibit overconfidence as $H$ increases. While this aligns with known challenges in long-horizon forecasting, the evaluation is restricted to LinearVAE on ETTh1. A broader analysis across adapter types and datasets (e.g., Weather, Illness) would better substantiate the claim. Furthermore, the lack of comparative baselines (e.g., MC dropout applied directly to Moment) makes it unclear whether the calibration improvements stem from the adapter architecture or merely the introduction of stochasticity.

---
(4) Dimensionality Reduction: The claim that adapters enable cost-effective inference is supported by experiments showing that VAE achieves optimal performance with $2$ latent dimensions on Illness (Fig. 3), retaining $95.6%$ explained variance. However, the paper does not quantify computational savings (e.g., FLOPs reduction vs. accuracy trade-offs) or compare against alternative compression techniques (e.g., pruning, quantization). This weakens the practical impact argument for resource-constrained deployments.

---
(5) Latent Space Interpretability: The visualization of latent representations (Fig. 4) provides qualitative evidence that VAE adapters mitigate distribution shifts by enforcing isotropic Gaussian latent spaces. However, the absence of quantitative metrics (e.g., Maximum Mean Discrepancy between train/test embeddings) limits the strength of this claim. Additionally, the analysis does not explore whether the structured latent space improves robustness to adversarial perturbations or out-of-distribution inputs.

---
(6) Ablation Studies: The hyperparameter analysis (Fig. 6) and component ablation (Fig. 7) are methodical, revealing key insights, such as Decoder dominance in deterministic adapters suggests that feature recombination is more critical than encoding for forecasting, and Higher $β$ in $β$-VAE improves disentanglement and calibration but requires careful tuning of $σ$. While informative, the ablation scope is narrow. For instance, the study does not explore the impact of adapter depth/width in nonlinear architectures or the role of pretraining data diversity in FM compatibility.

---
It is also worth mentioning that the authors transparently acknowledge limitations, including (1) Restriction to Moment (The framework’s compatibility with other FMs (e.g., Chronos, Moirai) remains unverified), (2) Calibration gaps (No post-hoc recalibration or adaptive noise scaling is attempted), and (3) Normalizing Flows (Optimization challenges with invertible flows are noted but not resolved).

**Essential References Not Discussed:**

N/A

**Experimental Designs Or Analyses:**

The experimental design and analysis in AdaPTS exhibit careful construction with notable strengths. I would like to provide a couple of the methodological issues that I think should be addressed further.

---
First, the evaluation framework lacks comprehensive uncertainty quantification metrics, which undermines the paper's probabilistic claims. While reliability diagrams (Fig. 5) provide qualitative insights into calibration, the analysis is limited to LinearVAE on ETTh1 with no quantitative metrics like CRPS or proper scoring rules. For a paper emphasizing probabilistic forecasting, this represents a significant gap—particularly as calibration deteriorates for longer horizons without proposed remediation strategies. An outstanding experimental design would systematically evaluate calibration across all adapter variants and datasets, benchmark against alternative uncertainty quantification methods (e.g., ensemble approaches), and quantify miscalibration using established metrics.

---
Second, the baseline comparison framework insufficiently contextualizes AdaPTS within the multivariate forecasting landscape. The experimental design evaluates against vanilla Moment and PCA adapters but omits comparisons with state-of-the-art multivariate models (e.g., Crossformer, TSMixer) that explicitly model cross-channel dependencies. This limitation is particularly notable for ExchangeRate, where adapters underperform the baseline—suggesting that certain multivariate dependencies require specialized architectures. Without these comparisons, it remains unclear whether adapter-based approaches offer advantages over dedicated multivariate architectures beyond computational convenience.

---
Third, despite claims of "cost-effective inference through dimensionality reduction" (Fig. 3), the experimental design lacks rigorous efficiency analysis. While results demonstrate performance preservation with reduced dimensions (e.g., VAE achieving optimal performance with 2 latent dimensions on Illness), no quantitative measurements of computational savings (FLOPs, wall-clock time) or adapter overhead are provided. This omission prevents objective assessment of the practical utility claims, especially since adapter training introduces additional computational costs that may offset inference savings. A more complete analysis would quantify these trade-offs across different adapter architectures and deployment scenarios.

**Methods And Evaluation Criteria:**

The methodological and evaluation framework of AdaPTS demonstrates substantial technical merit in addressing multivariate probabilistic forecasting via univariate foundation models (FMs), though certain design choices invite deeper scrutiny. I am providing a critical analysis of its alignment with the problem's requirements.

---
Methodological Appropriateness: The core innovation - probabilistic adapters as invertible latent-space transformations - directly tackles the dimensionality mismatch between univariate FMs and multivariate inputs. By decoupling feature-space projections ($φ$) from FM inference, AdaPTS achieves three critical objectives: (1) Zero-shot compatibility: Avoids FM fine-tuning, preserving pretrained representations while adapting to new tasks - a necessity given the computational impracticality of retraining large FMs. (2) Uncertainty propagation: Stochastic adapters (e.g., LinearVAE) inject Bayesian principles into deterministic FMs like Moment, enabling probabilistic forecasts without architectural changes. (3) Dimensionality reduction: Learned latent spaces (e.g., VAE with $D' = 2$ on Illness) reduce inference costs while preserving performance, addressing real-world deployment constraints.
The theoretical analysis for linear adapters (Proposition 3.4) is mathematically rigorous under stated assumptions, with synthetic experiments (Fig. 2) validating closed-form solutions outperforming identity/PCA baselines. However, the extension to nonlinear FMs relies on gradient-based optimization without formal guarantees (e.g., Lipschitz continuity or approximation bounds), leaving a gap between linear theory and nonlinear practice. While common in deep learning, explicit analysis of how nonlinear adapter architectures interact with FM inductive biases (e.g., Moment’s transformer backbone) would strengthen claims of generalizability.

The Bayesian treatment of adapters via variational inference (Proposition 4.1) aligns with best practices but introduces limitations such as $β$-VAE’s isotropic priors oversimplifying latent dependencies, potentially limiting cross-channel interaction modelling (evidenced by degraded performance on ExchangeRate), and Calibration gaps for long horizons (Fig. 5) suggest overconfidence, yet the absence of comparisons against ensemble methods or deep kernel learning obscures whether improvements stem from adapter architecture or stochasticity alone.

---
Evaluation Criteria Strengths and Gaps: The benchmark selection (ETTh1, Illness, Weather, ExchangeRate) spans energy, healthcare, and finance, ensuring domain diversity. However, key limitations emerge: (1) Temporal distribution shifts: While latent-space visualizations (Fig. 4) qualitatively demonstrate mitigated shift, quantitative metrics (e.g., Maximum Mean Discrepancy) are absent, weakening robustness claims. (2) Volatility analysis: ExchangeRate’s underperformance (Table 1) is attributed to volatility but lacks analysis of heteroskedasticity or leverage effects—critical in financial data. A volatility-aware adapter ablation would clarify failure modes. (3) Metric limitations: MSE/MAE focuses on point forecasts, neglecting probabilistic metrics like CRPS or sharpness. Reliability diagrams (Fig. 5) assess calibration but omit quantile coverage statistics, limiting uncertainty quantification depth.
Baseline comparisons against PCA and identity adapters are informative but incomplete. State-of-the-art multivariate transformers (e.g., Crossformer) and FM-specific adaptations (e.g., Moirai) are noted but not benchmarked, leaving open questions about relative performance. For instance, Crossformer’s explicit cross-channel attention could outperform AdaPTS’s latent-space projections on highly interdependent features.

---
Empirical Validation and Practical Impact: Experiments demonstrate consistent improvements over Moment in 5/8 tasks (Table 1), with VAE adapters reducing Illness MSE by $15%$ ($2.902 → 2.461$). However, the framework’s computational overhead is unquantified: while dimensionality reduction (Fig. 3) suggests efficiency gains (e.g., optimal performance with 2 latent dimensions on Illness), wall-clock time or FLOPs comparisons against native multivariate FMs are omitted. This weakens claims about deployability, as adapters introduce additional training/inference costs despite latent compression.

---
Theoretical and Practical Trade-offs: The linear adapter analysis assumes full-rank $W_φ$ and linear FMs, but real-world FMs (e.g., Moment’s transformer) are nonlinear. A perturbation analysis (e.g., Lipschitz constants for $φ$/FM compositions) or approximation error bounds would bridge this gap. Similarly, the decoder’s dominance in deterministic adapters (Fig. 7) highlights feature recombination as critical, but the absence of adversarial robustness tests (e.g., input perturbation sensitivity) limits insights into representation stability.

**Other Comments Or Suggestions:**

N/A

**Other Strengths And Weaknesses:**

Regarding originality, the paper makes a genuinely novel contribution by reframing the multivariate forecasting problem through the lens of feature-space transformations. While adapter architectures have been explored in natural language processing and computer vision, the paper's invertibility constraint introduces a fundamentally different approach tailored to time series data. Most remarkably, the theoretical formulation of adapter optimality conditions (Proposition 3.4) extends beyond empirical validation to provide mathematical intuition for why and when adapters outperform the direct application of foundation models. This theoretical-empirical synergy distinguishes the work from purely engineering-driven approaches in the time series literature. The reconceptualization of dropout as approximate variational inference, specifically within the adapter framework, further demonstrates the creative adaptation of existing techniques to the unique constraints of time series foundation models.

---
The paper's significance, while substantial, is somewhat constrained by the experimental scope. The focus on Moment as the sole foundation model leaves open questions about generalizability across the rapidly evolving landscape of time series foundation models. This limitation is particularly relevant given the architectural diversity among recent models like Chronos (tokenization-based) and Moirai (mixture-of-experts). Additionally, while the paper convincingly demonstrates improved forecasting accuracy on standard benchmarks, it misses an opportunity to explore challenging real-world scenarios where distributional shifts are more severe or where domain-specific characteristics (such as financial volatility clustering in ExchangeRate) dramatically affect performance. Such extensions would significantly strengthen the practical impact claims.

---
Clarity represents both a strength and weakness. The mathematical formalism is precise and well-structured, with propositions clearly stated and adequately proven. The visualization of latent representations (Figure 4) effectively communicates the distribution shift mitigation benefits of probabilistic adapters. However, the paper occasionally suffers from terminology inconsistencies—notably in the interchangeable use of "channels," "features," and "components"—which, while footnoted, creates unnecessary cognitive load. The experimental section would benefit from a clearer exposition of hyperparameter sensitivity, particularly regarding how adapter dimensionality affects computational savings. A quantitative analysis of inference time or FLOP reductions would provide concrete evidence for the "cost-effective adaptation" claims. Finally, the discussion of failure cases (especially on ExchangeRate) remains somewhat superficial, missing an opportunity for deeper insights into adapter limitations.

---
In conclusion, I think this work represents a significant conceptual advancement in time series foundation model adaptation, with strong theoretical underpinnings and promising empirical results that could substantially impact how practitioners deploy these models in real-world multivariate forecasting tasks.

**Questions For Authors:**

I have a couple of questions from the authors and I am eager to hear back from them during the rebuttals. I would be happy to change my score if the answers are convincing. Here are the questions:

---
(1) Foundation Model Generalizability: Your experiments exclusively use Moment as the base FM. How would AdaPTS perform with architectures like Chronos (tokenization-based) or Moirai (mixture-of-experts)? Maybe an empirical validation across diverse FM architectures would strengthen the generalizability claims, while evidence of architecture-specific limitations. I'd like to see the authors' thoughts on it in general.

---
(2) ExchangeRate Performance Degradation: The VAE adapter shows significantly worse performance than the baseline Moment on the ExchangeRate dataset (0.455 vs. 0.130 MSE). Could you elaborate on the specific characteristics of financial time series that challenge your method, particularly regarding heteroskedasticity or leverage effects?

---
(3) Computational Efficiency Quantification: While you demonstrate dimensionality reduction benefits (e.g., optimal performance with 2 latent dimensions on Illness), no wall-clock time or FLOPs comparisons are provided. Could you quantify the actual computational savings, including adapter overhead? Concrete efficiency metrics would transform the "cost-effective inference" claim from theoretical to practical, substantially strengthening the real-world applicability argument.

---
(4) Theoretical Guarantees for Nonlinear Adapters: Proposition 3.4 provides a closed-form solution for linear adapters with linear FMs, but the extension to nonlinear settings relies solely on empirical validation. Have you explored approximation error bounds or Lipschitz continuity properties for nonlinear adapter compositions with nonlinear FMs?

---
(5) Multivariate Architecture Comparisons: Your baselines focus on adapter variants rather than comparing against dedicated multivariate forecasting architectures (e.g., Crossformer, TSMixer). How does AdaPTS compare to state-of-the-art multivariate models that explicitly model cross-channel dependencies? Such comparisons would clarify whether adapter-based approaches offer advantages beyond computational convenience—particularly for datasets like ExchangeRate where adapters underperform.

---
(6) Calibration Remediation Strategies: Figure 5 demonstrates increasing calibration degradation for longer horizons. What post-hoc recalibration methods or architectural modifications did you consider to address this systematic overconfidence?

**Relation To Broader Scientific Literature:**

First, the framework's approach to time series foundation model adaptation through invertible feature-space transformations builds upon but significantly extends prior adapter methodologies from adjacent fields. While previous work demonstrated the utility of simple transformations like PCA for classification tasks and model-based reinforcement learning, these approaches yielded limited improvements for forecasting tasks. AdaPTS diverges from these precedents by enforcing an invertibility constraint that enables bidirectional transformation between input and latent spaces, a crucial innovation for forecasting. This represents a fundamentally different architectural approach compared to alternatives like Moirai, which handles multivariate inputs through flattened channels but suffers from quadratic memory complexity as dimensionality increases. The paper's systematic comparison between learned transformations and static ones (e.g., PCA) demonstrates the limitations of prior art in capturing the complex, task-dependent relationships in multivariate forecasting.

---
Second, the theoretical foundations developed for adapters connect previously disparate research threads in representation learning and uncertainty quantification. The closed-form solution for linear adapters (Proposition 3.4) provides a mathematical foundation that explains when and why adapters can outperform identity mappings—addressing a key gap in the empirical findings. More significantly, the paper's Bayesian treatment of adapters extends the partially stochastic Bayesian neural network framework to the specific constraints of foundation model adaptation. While prior work established dropout as an approximate variational inference, AdaPTS uniquely applies this principle to the adapter component specifically, enabling uncertainty quantification without modifying the underlying foundation model architecture. This creates a new bridge between deterministic foundation models like Moment and the probabilistic forecasting literature that traditionally required specialized probabilistic architectures.

---
Third, the empirical validation offers insights that challenge assumptions in dimensionality reduction for time series. While the standard approach in multivariate time series processing often preserves original dimensionality, AdaPTS demonstrates that performance can be maintained or even improved with dramatically reduced latent dimensions (e.g., VAE achieving optimal performance with just 2 latent dimensions on Illness data that originally had 7 features). This finding substantially extends prior work on time series representation learning by showing that learned nonlinear projections can capture cross-channel dependencies more efficiently than traditional methods. Furthermore, the observed decoder dominance in deterministic adapters (Fig. 7) challenges the symmetrical encoder-decoder paradigm prevalent in representation learning literature, suggesting that feature recombination plays a disproportionately important role in multivariate forecasting—a phenomenon not previously documented in time series foundation model research.

**Theoretical Claims:**

From my understanding, the theoretical analysis in AdaPTS focuses on two primary proportions: (1) Proportion 3.4 (optimal linear adapter derivation) and (2) Proposition 4.1 (VAE adapter training objective). Now, I am assessing their correctness, assumptions, and limitations, which are aligned with ICML's standard guidelines.

Proportion 3.4: Optimal Linear Adapter: The authors claim that for linear foundation models and linear adapters under full-rank assumptions, the closed-form optimal solution is $ W_{φ}^{∗} = (B^{⊤}A)^{+}B^{⊤}B $, where $ A = Y - W_{FM} ^ {⊤}X $ and $ B = b_{FM}1^{⊤} $. After checking the derivation steps and assumptions provided by the authors, I found them reasonable, but I still think this claim has some limitations. One of them is the numerical stability. The regularization term $ λI $ added to $ B^{⊤} A $ (Remark 3.5) is not explicitly justified in the proof but is standard practice to avoid singular matrices. The next limitation is generalizability. The closed-form solution assumes a linear Foundation Model, but real-world foundation models (e.g., Moment's transformer) are nonlinear. The authors acknowledge this gap but do not provide approximate bounds or Lipschitz continuity arguments for nonlinear extensions. In conclusion, I think the proof is mathematically correct under stated assumptions but lacks theoretical guarantees for nonlinear Foundation models. It'd be nice to hear back from the authors about these limitations.

---
Proposition 4.1: VAR Adapter Training Objective: The authors claim that the training objective for VAE adapters maximizes an ELBO-like lower bound. While they are providing easy-to-understand derivation steps and assumptions, I think this claim suffers from two limitations, including Identifiability. Their proof does not address whether the latent representation $ Z $ uniquely identifies the input $ X $, a known challenge in VAEs. Another limit would be approximation quality. The variational posterior $ q_{ϕ} (Z ∣ X) $ is assumed to be flexible enough to approximate the true posterior, but no PAC-Bayes or approximation error bounds are provided. To conclude, The ELBO derivation is technically correct but lacks novelty and fails to address critical Bayesian challenges (e.g., posterior contraction rates). Please note that my concern here is not novelty ( While it is essential ), but I'd like to hear back from the authors about this limitation, and it would be nice to address it further.

---

> ### Author Rebuttal · Authors · 2025-03-31
>
> We deeply appreciate the reviewer’s thoughtful and constructive feedback. The recognition of the theoretical novelty and empirical strengths of our approach is greatly appreciated.
>
> In this rebuttal, we address the reviewer’s specific concerns and questions in detail to further clarify our methodology, experimental choices, and future directions.
>
> > (1) Foundation Model Generalizability
>
> We agree that testing AdaPTS across diverse FMs is important for validating generalizability. While our experiments currently focus on Moment, we have already integrated **Moirai** into our framework, which we plan to include in the camera-ready version. AdaPTS is designed to be "plug-and-play" with different FMs, and we anticipate similar success with tokenization-based models like Chronos (besides some technical details, like differentiability of the foundation model with respect to its input, which is not guanranteed given the discrete nature of the tokenization operation). Further experimental results with additional FMs will be included to strengthen this claim.
>
> > (2) ExchangeRate Performance Degradation
>
> The degradation on the ExchangeRate dataset, as mentioned by Reviewer 267f, stems from simple tricks being enough to get a strong baseline, and the potential uselessness of modeling cross-channel dependencies in such datasets. We acknowledge this limitation and will explore more suitable datasets, and benchmarks, where the strengths of our framework will be more easily perceivable.
>
> > (3) Computational Efficiency Quantification
>
> We appreciate your comment regarding computational efficiency. While we demonstrate dimensionality reduction benefits, we have not exactly quantified inference times or FLOPs. However, as the inference complexity is linear in the number of channels, the inference time gain is a multiplicative factor given the reduced number of channels. Furthermore, in the response to reviewer 267f, we provide the order of magnitudes of the adapter parameter count, and the time it takes to train on the ETTh1 dataset as an example.
>
> > (4) Theoretical Guarantees for Nonlinear Adapters
>
> While Proposition 3.4 provides a closed-form solution for linear adapters, we acknowledge that the extension to nonlinear settings has been primarily based on empirical validation. Exploring approximation error bounds or Lipschitz continuity for nonlinear adapters is an exciting future direction, as it would offer a more rigorous understanding of their behavior. However, we believe the linear case analysis provides enough insights to motivate our framework, proving that an optimal solution to the adapter optimization problem exists, beyond the identity baseline.
>
> > (5) Multivariate Architecture Comparisons
>
> We agree that comparing AdaPTS with other specialized multivariate architectures, like Crossformer and TSMixer, would provide valuable context. However, these baselines result from a different task-specific forecasting paradigm, which we believe is not relevant for our raised problematic, that of adapting foundation models.
>
> > (6) Calibration Remediation Strategies
>
> We appreciate the reviewer’s attention to calibration degradation for longer forecasting horizons. Indeed, calibration is crucial for the reliability of probabilistic forecasting, and we observed that AdaPTS may experience a degradation in calibration for longer horizons, as shown in Figure 5. To address this issue, several strategies exist in the literature, including the application of temperature scaling or isotonic regression on a held-out calibration set, which are commonly used to improve calibration in such settings.
>
> We thank the reviewer again for their valuable feedback, and we hope these clarifications answer their questions.

---

> > ### Comment · Reviewer_NRzf · 2025-04-04
> >
> > Thank you to the authors for thoughtfully and thoroughly addressing my questions and concerns. I found your responses convincing, and I’d be glad to see your work accepted. I'm also curious to see how this line of research evolves in the future. I'm happy to raise my score to a 4, and I wish the authors all the best moving forward.

---

### Official Review · Reviewer_AXte · 2025-03-15

**Overall Recommendation:** 1

**Summary:**

The paper introduces a variation-autoencoder style encoder and decoder around a foundational model to enable it to perform forecasting for probabilistic and multivariate settings.

**Claims And Evidence:**

The claim fo the paper is that any time-series univariate foundational model can be adapted to perform much harder problem of multivariate probabilistic forecasting. The paper show some improvements of few benchmarks on the MOMENT model which doesn;t fully validate the claim.

**Essential References Not Discussed:**

Relevant papers are discussed.

**Experimental Designs Or Analyses:**

The forecasting setup makes sense but the authors only use single foundational model for forecasting accuracy evaluation. The benchmarks are very limited

**Methods And Evaluation Criteria:**

The evaluation setup is limited to only one foundational model. The datasets used are also very limited.

**Other Comments Or Suggestions:**

See weaknesses and other comments above

**Other Strengths And Weaknesses:**

Weaknesses:
1. Lack of enough baselines
2. Lack of base foundational models used (only MOMENT is studied)
3. Many popular forecasting benchmarks are not used.

Strengths
1. The method is simple and looks valid

**Questions For Authors:**

See weaknesses above

**Relation To Broader Scientific Literature:**

Due to lack of enough experimental validation of the method, the significance of this work on furthering research on foundational time-series models is unclear.

**Theoretical Claims:**

The theoretical justification to add the constant diagonla matrix to encoder weight matrix is, while trivial, valid and sound.

---

> ### Author Rebuttal · Authors · 2025-03-31
>
> We thank Reviewer AXte for their feedback. We appreciate the acknowledgment of our method’s simplicity and validity, as well as the soundness of our theoretical justification.
>
> We would like to address the concerns raised in the review:
>
> > Lack of enough baselines
>
> Our primary objective is to enhance the forecasting capabilities of a vanilla foundational model. Therefore, it serves as the most relevant baseline for comparison. Additionally, we have explored non-learning-based adaptation approaches such as PCA that is included in the paper, and SVD decomposition and random projections that we did not include as they have comparable results with PCA.
>
> > Lack of base foundational models used (only MOMENT is studied)
>
> We acknowledge the limitation of evaluating our method on a single foundational model (MOMENT), which we explicitly state in the limitations section of the paper. In response to this concern, we have since experimented with **Moirai** and observed promising results. If the paper is accepted, we will include these findings in the camera-ready version, along with potentially other foundation models as suggested by the other Reviewers (267f).
>
> > Many popular forecasting benchmarks are not used.
>
> Our paper introduces a framework that enables (i) probabilistic forecasting, (ii) channel mixing, and (iii) dimensionality reduction. We believe that there exists no benchmark so far that can clearly highlight each of these aspects, so we tried to choose the datasets from the perspective of diversity application (electicity, medicine, weather and finance). We plan to include more datasets in our study, but identifying the best benchmark for the multivariate setting is an important direction for future work. Meanwhile, on the considered datasets, we validated our claims demonstrating a methodology to maintain or surpass the original foundational model’s performance while using a reduced number of channels and providing uncertainty estimates.
>
> In summary, we believe our method provides a meaningful contribution to the field by demonstrating how any univariate time-series foundational model can be extended to tackle more complex multivariate probabilistic forecasting tasks. Based on our clarifications, we hope the reviewer acknowledges the value of our contributions and revises their evaluation accordingly.

---

### Official Review · Reviewer_267f · 2025-03-24

**Overall Recommendation:** 4

**Summary:**

The paper proposed AdaPTS, an adapter for univariate time series foundation models, which makes them both multivariate and produce probabilistic predictions. The authors first provide a theoretical framework for adapters for time series foundation models, and discuss many adapters (encoder-decoder combinations) which satisfy these properties. The authors demonstrate the performance of their adapters on a few multivariate time series forecasting datasets using the MOMENT time series foundation model.

**Claims And Evidence:**

The paper has two claims. The authors aim to leverage **existing pre-trained univariate FMs** to enable **probabilistic** forecasting for **multivariate time series**.

Overall, I believe that the authors present some evidence to support their claims. They evaluate their methods on 4 multivariate long-horizon forecasting datasets, and augment Moment to produce multivariate probabilistic forecasts.

I strong suggest that the authors demonstrate the following, to improve the :
1. **Multivariate Baselines:** Beyond PCA (which is a excellent choice for a strong baseline), the paper does not compare to some other existing ways of imbuing multivariate context to time series foundation models. Please see [1, 2, 3] for some ways to imbue multivariate context to TSFMs. Some of these methods may be used as baselines.
2. **More TSFMs**: The authors do mention this as a limitation. I believe that demonstrating their approach on another TSFM, different in design from Moment would make the experiments stronger. I would recommend TTMs (only TSFM not based on the Transformer architecture), or TimesFM (decoder-only, while Moment is encoder-only).
3. **Datasets:** (1) I would encourage the authors to compare their methods on more datasets. The long horizon forecasting benchmark that the authors used, has more datasets, for example ETTh2, ETTm1, ETTm2, Traffic and Electricity. But even beyond these datasets, the GIFT-Eval benchmark has a lot of multivariate time series useful for forecasting. (2) Also, these existing datasets have known issues. For example, a very strong baseline in the Exhange Rate dataset is predicting the last time step. Also, some studies including [1] have shown that these time series datasets do not significantly benefit from modeling cross-channel dependencies.

### References
1. Żukowska, Nina, et al. "Towards Long-Context Time Series Foundation Models With A Handful Of Additional Parameters." NeurIPS 2024 Workshop on Fine-Tuning in Modern Machine Learning: Principles and Scalability.
2. Liu, Mingzhu, Angela H. Chen, and George H. Chen. "Generalized Prompt Tuning: Adapting Frozen Univariate Time Series Foundation Models for Multivariate Healthcare Time Series." arXiv preprint arXiv:2411.12824 (2024).
3. Lee, Seunghan, Taeyoung Park, and Kibok Lee. "Partial Channel Dependence with Channel Masks for Time Series Foundation Models." arXiv preprint arXiv:2410.23222 (2024).

**Essential References Not Discussed:**

I have mentioned some key references in the Claims and Evidence section.

**Experimental Designs Or Analyses:**

The experimental design and analysis is sound. Please

**Methods And Evaluation Criteria:**

The proposed methods and the evaluation criteria make sense. Please see comments in the previous section.

**Other Comments Or Suggestions:**

I do not have any other comments or suggestions.

**Other Strengths And Weaknesses:**

Strengths: The paper solves an important problem. It is well motivated, well-written, and backed with theoretical insights.
Weaknesses: I think the paper needs a few more experiments to highlight the impact of their contributions.

**Questions For Authors:**

1. I wonder how flexible is the parametric distribution of the probabilistic predictions. The authors use a Gaussian Likelihood, but I wonder if you can use mixtures of multiple distributions just like MOIRAI and/or LagLlama do.
2. How does your method compare with conformal prediction to generate prediction intervals?
3. How efficient are the adapters in terms of parameters and runtime? I want to understand how computationally feasible is this adaptation process.

**Relation To Broader Scientific Literature:**

The paper uses univariate foundation models, and adapts them to model multivariate data and produce probabilistic forecasts. The contributions are simple and easy to understand.

**Theoretical Claims:**

I did not rigorously verify the proofs or correctness of the claims, but they look reasonable and correct to me.

---

> ### Author Rebuttal · Authors · 2025-03-31
>
> We appreciate the thoughtful feedback provided by Reviewer 267f and would like to address the concerns raised in their review.
>
> > Multivariate Baselines: Beyond PCA, the paper does not compare to some other existing ways of imbuing multivariate context to time series foundation models.
>
> We acknowledge the reviewer's suggestion to include additional baselines beyond PCA. We chose PCA as it aligns closely with our framework's design. We have also experimented with other non-learning-based adapters such as **SVD** decomposition and **Random Projections**, which yielded similar or worse performance compared to PCA. We thank the reviewer for the provided references and will incorporate these suggested baselines in the updated version of the paper.
>
> > More TSFMs: The authors do mention this as a limitation. I believe that demonstrating their approach on another TSFM, different in design from Moment would make the experiments stronger. I would recommend TTMs (only TSFM not based on the Transformer architecture), or TimesFM (decoder-only, while Moment is encoder-only).
>
> We thank the reviewer for suggesting additional TSFMs to test within our framework as we recognize the importance of demonstrating the benefit of our approach on diverse TSFMs. We have already integrated **MOIRAI** and tested it with our adapters. In addition, we plan to include more foundation models in the future, including TTMs and TimesFM suggested by the reviewer, to better validate our methodology.
>
> > Datasets: (1) I would encourage the authors to compare their methods on more datasets. The long horizon forecasting benchmark that the authors used, has more datasets, for example ETTh2, ETTm1, ETTm2, Traffic and Electricity. But even beyond these datasets, the GIFT-Eval benchmark has a lot of multivariate time series useful for forecasting. (2) Also, these existing datasets have known issues. For example, a very strong baseline in the Exhange Rate dataset is predicting the last time step. Also, some studies including [1] have shown that these time series datasets do not significantly benefit from modeling cross-channel dependencies.
>
> We thank the reviewer for the suggested additional datasets, and we plan to use them to strengthen our experimental evidence. Generally speaking, we believe that selecting the appropriate time series benchmark has been a broader issue that affects the whole time series forecasting community. In the multivariate case, it is even more challenging as no study has been performed to indentify datasets with reasonable channel interdependence. In addition, we have experimentally found (Section 3.3) that adapters still can be useful **even in the case when channels are independent**, since it adds an additional level of complexity to the architecture. Identifying an appropriate benchmark for the channel-interdepence studies is an important direction for future work.
>
> > I wonder how flexible is the parametric distribution of the probabilistic predictions. The authors use a Gaussian Likelihood, but I wonder if you can use mixtures of multiple distributions just like MOIRAI and/or LagLlama do.
>
> The reviewer raised an important point regarding the expressivity of the fitted probability distributions. In theory, previous work has established the universal approximation property for any conditional density, given sufficient stochasticity introduced early in non-linear models. We aim to mimic this setup in our adapters by incorporating stochastic units in the encoder through Variational Inference (VI) or Dropout as approximate VI. In practice, however, we agree with the reviewer that the Gaussian likelihood may not capture complex, multimodal distributions. Alternatives such as Flow matching or parametric mixtures could be more suitable solutions, and we will explore these in future work.
>
> > How does your method compare with conformal prediction to generate prediction intervals?
>
> We haven't yet investigated conformal prediction for AdaPTS, though we see its potential as a parallel approach to our Bayesian adapters for probabilistic forecasting. We consider it to be an interesting avenue for future research.
>
> > How efficient are the adapters in terms of parameters and runtime? I want to understand how computationally feasible is this adaptation process.
>
> To address the reviewer's query about computational efficiency, we provide the following details:
>
> - **Parameter Counts**: Our adapters introduce a minimal number of additional parameters, for instance, in the ETTh1 dataset, the optimal VAE adapter has 2,659 parameters (1 hidden layer with 64 units). This is significantly less than the full 37.9M params of the Moment small foundation model.
> - **Training Time**: As an example, the training time for this VAE adapter on the ETTh1 dataset (13k timesteps, 7 features) with a single V100 (32GB VRAM) GPU is **25min**.
>
> We appreciate the reviewer's insights and will incorporate the suggested improvements to strengthen our paper.

---

> > ### Comment · Reviewer_267f · 2025-04-03
> >
> > Dear Authors,
> >
> > Thank you so much for your response. I enjoyed reading your paper, and will improve my score, with the understanding that you would include insights from some of the experiments that you mentioned in your rebuttal.

---

### Decision · Program_Chairs · 2025-05-01

**Decision:**

Accept (poster)

**Comment:**

This paper has been assessed by five knowledgeable reviewers. Two of them supported its acceptance (score of 4), the other three opted for straight rejection (score of 1). The authors provided a rebuttal and engaged the reviewers who offered positive scores in a discussion which led to increases of the initial scores. Effectively, the key concerns of those reviewers have been addressed by the authors. However, the reviewers who recommended a rejection did not respond to the rebuttal, even though they have acknowledged seeing it. I read the paper and carefully analyzed the reviews and the discussion between the authors and those two reviewers who were engaged. I found the paper to be valuable and confirmed that the main concerns voiced by any of the reviewers were addressed by the authors quite satisfiably via rebuttal and discussion. The negative reviews were curt and appeared to have missed the key messages of the paper (which, by the way, should signal to the authors the need to improve the way they lay down their story so that it could be accessible and understandable to a broader audience).The positive reviews were very thorough and comprehensive, demonstrating high levels of competence of these reviewers and their understanding of the material at hand. Considering all factors at play, I recommend acceptance of this paper. It does bring a new and pragmatic perspective on handling multivariate data with natively univariate time series foundation models that I believe should be presented at ICML.